# Synthetic microparticles conjugated with VEGF$_{165}$ improve the survival of endothelial progenitor cells via *microRNA-17* inhibition

Sezin Aday[1,2,3,4], Janet Zoldan[5], Marie Besnier[4], Laura Carreto[6], Jaimy Saif[4], Rui Fernandes[7], Tiago Santos[8], Liliana Bernardino[8], Robert Langer[3], Costanza Emanueli[4,9] & Lino Ferreira[1,2]

Several cell-based therapies are under pre-clinical and clinical evaluation for the treatment of ischemic diseases. Poor survival and vascular engraftment rates of transplanted cells force them to work mainly via time-limited paracrine actions. Although several approaches, including the use of soluble vascular endothelial growth factor (sVEGF)—VEGF$_{165}$, have been developed in the last 10 years to enhance cell survival, they showed limited efficacy. Here, we report a pro-survival approach based on VEGF-immobilized microparticles (VEGF-MPs). VEGF-MPs prolong VEGFR-2 and Akt phosphorylation in cord blood-derived late outgrowth endothelial progenitor cells (OEPCs). In vivo, OEPC aggregates containing VEGF-MPs show higher survival than those treated with sVEGF. Additionally, VEGF-MPs decrease *miR-17* expression in OEPCs, thus increasing the expression of its target genes *CDKN1A* and *ZNF652*. The therapeutic effect of OEPCs is improved in vivo by inhibiting *miR-17*. Overall, our data show an experimental approach to improve therapeutic efficacy of proangiogenic cells for the treatment of ischemic diseases.

[1] CNC-Center for Neuroscience and Cell Biology, University of Coimbra, 3004-517 Coimbra, Portugal. [2] Faculty of Medicine, University of Coimbra, 3000-548 Coimbra, Portugal. [3] Department of Chemical Engineering, Massachusetts Institute of Technology, Cambridge, MA 02139, USA. [4] Bristol Heart Institute, School of Clinical Sciences, University of Bristol, Bristol BS2 8HW, UK. [5] Department of Biomedical Engineering, Cockrell School of Engineering, The University of Texas at Austin, Austin, TX 78712, USA. [6] University of Aveiro, 3810-193 Aveiro, Portugal. [7] HEMS—Histology and Electron Microscopy Service, IBMC/I3S, Universidade do Porto, 4200-135 Porto, Portugal. [8] Health Sciences Research Centre, Faculty of Health Sciences, University of Beira Interior, 6201-506 Covilhã, Portugal. [9] National Heart and Lung Institute, Hammersmith Campus, Imperial College of London, London SW7 2AZ, UK. Correspondence and requests for materials should be addressed to C.E. (email: Costanza.Emanueli@bristol.ac.uk) or to L.F. (email: lino@uc-biotech.pt)

schemic diseases are leading cause of morbidity and mortality in the contemporary world. Several pre-clinical and clinical trials are exploring the therapeutic effect of cell-based therapies, in particular, bone marrow-derived proangiogenic cells and mesenchymal stem cells in ischemic diseases[1–3]. Unfortunately, most of the cells (more than 80%) die a few days (< 3 days) after delivery[4–6], thus hindering the therapeutic effect. Some approaches have been explored to augment cell survival in ischemic conditions. These include the exposure of transplanted cells to temperature shock, genetic modification of cells to overexpress growth factors and/or anti-apoptotic proteins and pre-conditioning the cells with pharmacological agents and cytokines[7, 8]. However, most of these methodologies have not reached the clinical trials, because they have shown limited effectiveness due to the multi-factorial nature of cell death. In addition, some of them are not cost-effective (e.g., recombinant proteins) or are difficult to implement from a regulatory stand-point (e.g., genetic modifications).

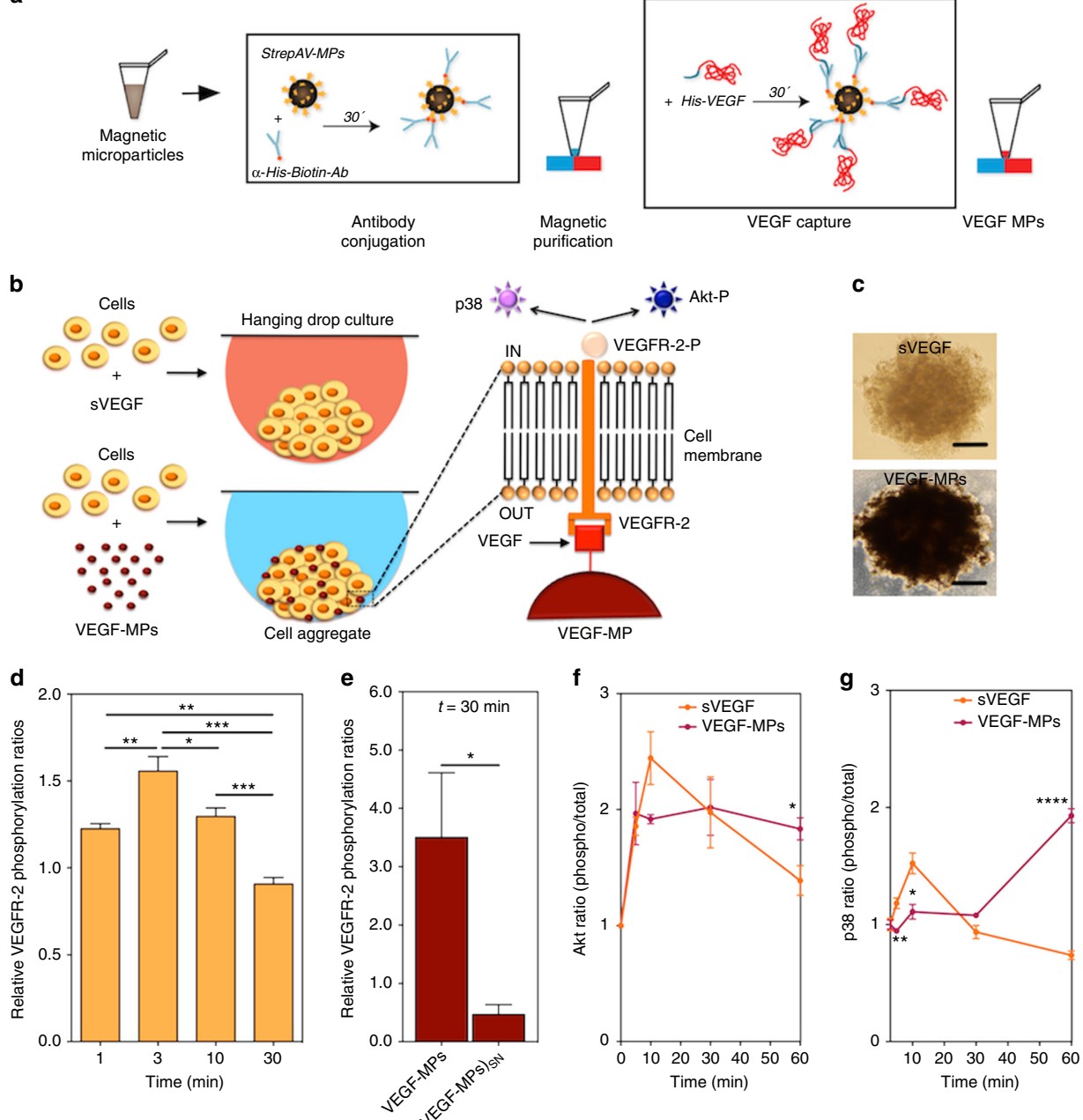

**Fig. 1** Preparation of VEGF-MPs and biological characterization in OEPCs. **a** Schematic representation of the protocol for the preparation of VEGF-conjugated particles and **b** for the formation of cell aggregates containing sVEGF or VEGF-MPs. **c** Light microscopy images of the OEPC aggregates at 24 h. The differences in color of cell aggregates are due to the presence of MPs. *Bar* corresponds to 50 μm. **d** VEGFR-2 phosphorylation in OEPC aggregates cultured in media containing sVEGF. **e** VEGFR-2 phosphorylation in OEPC aggregates containing VEGF-MPs or containing cell culture media exposed to the same number of MPs used to make the cell aggregates [(VEGF-MPs)$_{SN}$]. VEGF phosphorylation was quantified by ELISA. Values are given as average ± SEM (n = 4-8). **f**, **g** ELISA quantification of phospho-Akt/total Akt (**f**) and phospho-p38/total p38 (**g**). Values are given as average ± SEM (n = 3). In **d**, statistical analyses were performed using one-way ANOVA followed by a Bonferroni post test. In **e–g**, unpaired *t*-test was performed between groups. *$P \leq 0.05$, **$P \leq 0.01$, ***$P \leq 0.001$, and ****$P \leq 0.0001$

Vascular endothelial growth factor 165 (from now on referred as VEGF) is one of the most powerful and well-studied pro-survival/pro-angiogenic factors[9, 10]. Three main tyrosine kinase receptors initiate signal transduction cascades in response to soluble VEGF, including VEGF receptor 1 (VEGFR-1), VEGFR-2, and VEGFR-3[10, 11]. VEGF exerts its pro-survival effect mostly via VEGFR-2 by inducing its dimerization and subsequent phosphorylation of the receptor[12]. Although soluble and immobilized VEGF induce similar degrees of VEGF receptor phosphorylation, they have different properties[13]. In comparison to endothelial cells (ECs) cultured in the presence of soluble (free) VEGF (sVEGF), ECs cultured on VEGF-bound surfaces show different morphology[14], extended VEGFR-2 phosphorylation (mediated by the phosphorylation at the site Y1214 of the VEGFR-2 C-terminal tail)[13], and higher activation of the p38/MAPK pathway[13]. Due to the prolonged VEGFR-2 phosphorylation and higher activation of subsequent pathways, immobilized VEGF might be a promising pro-survival agent for cell-based therapies. However, in vitro and in vivo pro-survival effects of immobilized VEGF remain relatively elusive. In addition, the downstream molecular players mediating the biological effect of immobilized VEGF, particularly micro RNAs (miRNAs), are still unknown.

Herein, we have engineered microparticles (MPs) with VEGF and studied the pro-survival effect and function of VEGF-MPs in outgrowth endothelial progenitor cells (OEPCs). We have additionally investigated the involvement of miRNAs in the biological responses to immobilized VEGF. Previous studies on the differential signaling responses of immobilized VEGF focused on planar surfaces that are not transplantable in most cases, thus making the identification of in vivo molecular players of immobilized VEGF difficult[13, 15]. Here, we have immobilized VEGF onto magnetic MPs of 4.5 µm, which can be assembled with OEPCs in cell aggregates and are transplantable. In this study, we have used umbilical cord blood CD34$^+$ cell-derived OEPCs[3]. In contrast to the "early" endothelial progenitor cells, OEPCs directly participate in tubulogenesis[16] and their neovasculogenesis properties have been demonstrated in pre-clinical tests using different animal models of hindlimb ischemia, diabetic chronic wounds, among others[4, 5, 17–19]. Moreover, although OEPCs express endothelial cell markers such as vWF and VECAD; uptake acetylated LDL; and display the morphology of ECs, they show distinct features from mature ECs[6, 20].

We show that VEGF-MPs assembled into cell aggregates prolong the phosphorylation of VEGFR-2 and Akt in OEPCs in comparison to sVEGF. Furthermore, VEGF-MPs incorporated in cell aggregates increase OEPC survival both in vitro and in vivo as compared with cell aggregates, cell aggregates containing either uncoated MPs or sVEGF. We further show that the prolonged VEGFR-2 phosphorylation in cell aggregates containing VEGF-MPs is associated with the down-regulation of miR-17 and increase in the expression of its target genes CDKN1A and ZNF652.

The bioengineering platform used in this work opens an avenue to improve the mechanistic understanding of how VEGF immobilization alters cell behaviors and what are the molecular mediators of this process, thus enabling the design of previously undescribed therapies to treat ischemic diseases.

## Results

**VEGF can be immobilized onto magnetic MPs.** VEGF was immobilized onto magnetic MPs (4.5 µm diameter) using biotin-streptavidin chemistry (Fig. 1a). The biotin-streptavidin system is the strongest noncovalent biological interaction

known[21]. In this method, streptavidin-coated MPs were first conjugated with biotinylated anti-histidine (anti-his) antibody, and then, with histidine-tagged VEGF (his-VEGF). To determine the success of VEGF immobilization, MPs were separated by a magnet, washed (until no measurable leaching of VEGF was observed), and then exposed to a fluorescent secondary antibody against immobilized VEGF followed by flow cytometry characterization. Streptavidin-coated MPs conjugated with biotinylated anti-his antibody in the absence of his-VEGF showed no fluorescence (Supplementary Fig. 1a). In contrast, VEGF-MPs showed an increase in the mean fluorescence indicating that MPs were successfully conjugated with VEGF. Then, we evaluated different initial ratios of anti-his antibody and his-VEGF to maximize the concentration of immobilized VEGF. Depending on the initial concentrations of antibody and growth factor, the immobilized VEGF amounts were between $271.8 \pm 44.3$ and $425.4 \pm 50.2$ ng per $10^6$ microparticles (Supplementary Fig. 1b). For subsequent studies, MPs conjugated with $425.4 \pm 50.2$ ng VEGF per $10^6$ particles were used.

**MPs do not induce toxicity in OEPCs.** The toxicity of MPs against OEPCs was determined by cell proliferation, viability, and apoptosis assays. Firstly, iron release studies from the MPs were performed by inductively coupled plasma mass spectrometry (ICP-MS). It is known that MPs might have potential toxic effects especially due to iron release from them[22–24]. Polstyrene-coated MPs were used in our studies to prevent this iron release. Indeed, no release of Fe from MPs was determined up to 7 days in cell culture medium at 37 °C (Supplementary Fig. 1c). In order to evaluate the effect of MPs on cell proliferation, monolayers of OEPCs were used (Supplementary Fig. 2a). The cells were treated with different ratios of MPs (1:10–1:100; cell:MP ratios) up to 5 days and cell proliferation was quantified by a WST-1 assay. No measurable effect of MPs on cell proliferation was observed (Supplementary Fig. 2a). Then, cell viability was monitored in cell aggregates having different cell numbers and cell:MP ratios (Supplementary Fig. 2b, c). Cell aggregates with small number of cells ($\leq$10,000 cells) having a cell:MP ratio of 1:1 showed lower survival than cell aggregates without MPs (Supplementary Fig. 2b). Therefore, for subsequent experiments we have used cell aggregates with 20,000 cells having a diameter of approximately 150 µm (Fig. 1c). Then, we evaluated the cell:MP ratio. Cell aggregates with a cell:MP ratio of 1:1 seem to survive slightly better than the ones with a cell:MP ratio of 1:2 and thus were selected for subsequent experiments (Supplementary Fig. 2c). After selection of cell number and cell:MP ratio, we checked whether MPs increased apoptosis in cell aggregates. The MPs did not induce apoptosis in cell aggregates as determined by a TUNEL assay (Supplementary Fig. 2d, e).

**VEGF-MPs induce VEGFR-2 phosphorylation and increase $[Ca^{2+}]_i$.** The extracellular distribution of VEGFR-2 clusters was initially monitored to evaluate the effect of VEGF-MPs on VEGFR-2. VEGF-MPs induced VEGFR-2 clustering at the cell surface and the clustering was prolonged relatively to sVEGF (Supplementary Fig. 3a, b). To determine the activity of VEGF-MPs, we evaluated the phosphorylation level of VEGFR-2. OEPCs were starved for 20 h, harvested and suspended in basal medium containing either sVEGF or VEGF-MPs. In order to improve cell–MP interaction, the cell suspension was then seeded in hanging drops and incubated for different period of times (Fig. 1b, c). Cell aggregates without MPs or cell aggregates with uncoated MPs were used as controls. Cell aggregates treated with the same amount of sVEGF showed a rapid increase in the phosphorylation of VEGFR-2 (Fig. 1d) and returned to its basal

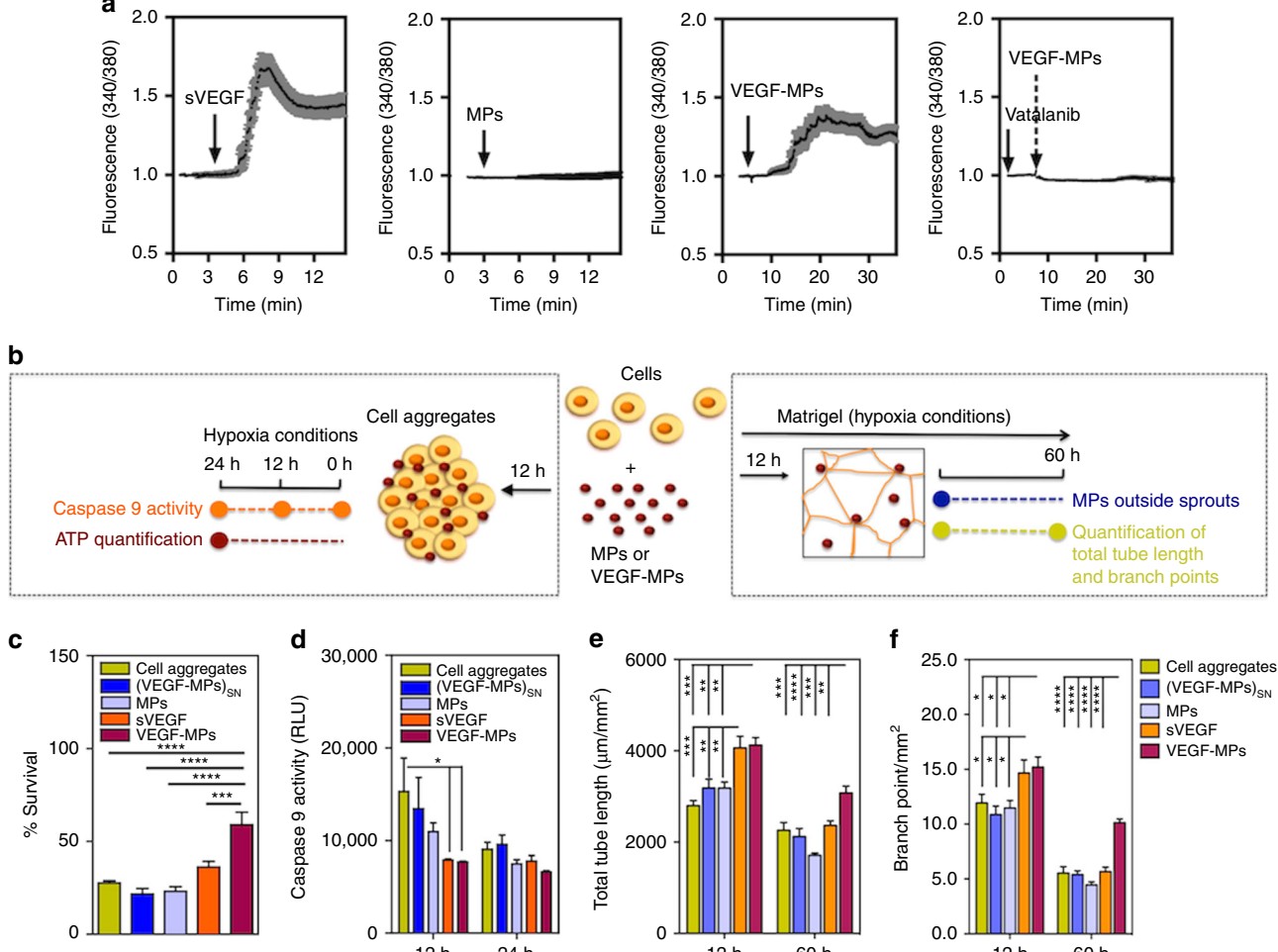

**Fig. 2** The biological effect of VEGF-MPs on OEPCs. **a** Single cell calcium measurements. OEPCs were starved in medium without serum for 20 h, loaded with a $Ca^{2+}$ probe and activated by VEGF, blank MPs, VEGF-MPs, or inhibited by Vatalanib, an inhibitor of VEGFR-2. The *arrows* indicate the time when the compounds or MPs were added. At least 10 cells have been monitored for intracellular $Ca^{2+}$ in each of the experimental groups. Averages and SEM values are in *black* and *grey*, respectively. **b** Schematic representation of the protocols used to demonstrate the higher OEPC survival and activity after exposure to VEGF-MPs than sVEGF. **c, d** The survival (**c** 24 h) and apoptosis (**d** 12 and 24 h) of OEPCs in aggregates under hypoxia in serum-deprived conditions as assessed by an ATP-based assay or the measurement of caspase 9 activity. **e, f** OEPC aggregates were cultured on Matrigel under hypoxia for 12 and 60 h, after which the tube length (**e**) and branching points (**f**) were measured. MPs indicate cell aggregates containing uncoated beads while (VEGF-MPs)$_{SN}$ indicates cell aggregates exposed to the supernatant of VEGF-MPs. In all graphs, values are given as average ± SEM ($n = 3$–8). Statistical analyses were performed using one-way ANOVA followed by a Bonferroni post test. $^*P \leq 0.05$, $^{**}P \leq 0.01$, $^{***}P \leq 0.001$, and $^{****}P \leq 0.0001$

level after 10 min. In contrast, cell aggregates with VEGF-MPs showed high levels of VEGFR-2 phosphorylation for at least 30 min (Fig. 1e).

To confirm that the phosphorylation of VEGFR-2 was mediated by the immobilized VEGF, but not VEGF leaching from the MPs, VEGF-MPs were suspended in cell culture medium at the same concentration used in cell aggregates and incubated at 37 °C for 30 min. After incubation, the MPs were removed and starved OEPCs were suspended in MP conditioned medium to generate hanging drops. No increase in VEGFR-2 phosphorylation was observed when VEGF-MP conditioned medium was used (Fig. 1e). These results confirm that the phosphorylation of VEGFR-2 in OEPC aggregates containing VEGF-MPs was mediated by the immobilized VEGF.

Phosphorylation and dephosphorylation of ligated or free VEGFR-2 occur on the cell surface and in the endosomes[25]. Computational models indicated that Rab4/5 endosomes contain more VEGFR-2 phosphorylated at the Y1175 site while the cell surface has more VEGFR-2 phosphorylated at the Y1214 site[25]. Previously, it was shown that VEGF conjugated onto flat surfaces

extended the phosphorylation of the VEGFR-2 at the Y1214 site and activated p38/MAPK pathway[13]. In our study, OEPCs internalize VEGF-MPs (Supplementary Figs. 3c and 4a), which may alter the phosphorylation profile of tyrosine residues in VEGFR-2. The phosphorylation of Y1175 and Y1214 peaked after 3–5 min for sVEGF treated group. A similar profile was observed for the phosphorylation of Y1214 in VEGF-MP-treated group, while the phosphorylation of Y1175 peaked at 10 min and maintained its levels for additional 50 min (Supplementary Fig. 5a, b). In line with Y1175 phosphorylation results, the phosphorylation of Akt was prolonged in VEGF-MP treated group relatively to sVEGF one (Fig. 1f). Finally, in agreement with previous studies, p38 phosphorylation was prolonged in VEGF-MP treated group than in sVEGF treated group[13] (Fig. 1g).

To further evaluate the bioactivity of VEGF-MPs, we assessed their capacity to increase intracellular free $Ca^{2+}$ via activation of VEGFR-2[26] (Fig. 2a). The phosphorylation of VEGFR-2 activates PLCγ, which in turn activates MAPK/ERK-1/2 pathway and also increases the intracellular levels of $Ca^{2+}$. In ECs, the initial increase in the cytosolic-free calcium is due to $Ca^{2+}$

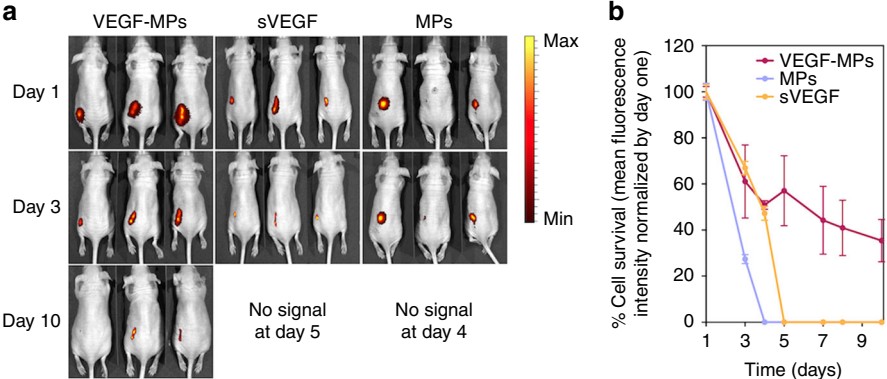

**Fig. 3** OEPC aggregates containing VEGF-MPs improve in vivo survival. OEPC aggregates were prepared with sVEGF, blank MPs or VEGF-MPs and implanted subcutaneously. **a** Representative IVIS images and **b** fluorescence intensity measurements of mice following injection of cell aggregates containing 1 million GFP-labeled OEPCs with sVEGF, blank MPs, or VEGF-MPs. The fluorescence intensities were normalized by day one. Results are average $\pm$ SEM ($n = 7$)

mobilization from intracellular stores and only afterwards is due to extracellular $Ca^{2+}$ influx[26]. To monitor intracellular levels of $Ca^{2+}$ on single cell level, OEPCs were starved in basal medium without supplements for 20 h and loaded with the calcium-sensitive fluorescent dye, Fura-2 AM. Firstly, the cells were activated by sVEGF (Fig. 2a). The administration of VEGF induced a fast increase in intracellular free $Ca^{2+}$ after 3 min, peaking at 4–5 min, followed by a decrease in the intracellular $Ca^{2+}$ (6–7 min) and a plateau phase where $Ca^{2+}$ remains constant for the remaining of the experiment (at least 9 min). The rapid response to sVEGF is due to the fast interaction of VEGF with its receptor and the internalization of the receptor-ligand complex[26]. In contrast to sVEGF, VEGF-MPs stimulated the OEPCs continuously for a longer time. In our experiments, changes were followed up to 45 min and an oscillation in $Ca^{2+}$ was observed for the cells stimulated with VEGF-MPs (Fig. 2a). Importantly, no changes in the intracellular $Ca^{2+}$ levels were observed for cells exposed to blank MPs. When VEGFR-2 inhibitor Vatalanib was used, the signal was abolished for the cells treated with VEGF-MPs.

Overall, immobilized VEGF, but not sVEGF, prolongs the phosphorylation of VEGFR-2 in OEPCs. This effect seems to be associated with a prolongation in the phosphorylation of tyrosine site Y1175 and the phosphorylation of Akt. In addition, immobilized VEGF prolongs the phosphorylation of p38 as previously reported[13]. The phosphorylation of VEGFR-2 by immobilized VEGF also induced a prolonged intracellular $Ca^{2+}$ increase that was abolished by chemically inhibiting VEGFR-2 with Vatalanib.

**VEGF-MPs enhance in vitro and in vivo survival of OEPCs.** Next, we evaluated whether VEGF-MPs could enhance the survival of OEPCs cultured under hypoxic conditions (0.5% $O_2$), to mimic the environment that cells encounter after transplantation in ischemic tissues. MPs were mixed with a suspension of OEPCs and cell aggregates were formed by a hanging drop methodology[27] (Fig. 2b). At 24 h, cell aggregates containing VEGF-MPs showed 60% higher ATP production (and thus indirectly cell viability) than cell aggregates treated with sVEGF (Fig. 2c). This effect was likely related to the prolonged activation of pro-survival Akt pathway in VEGF-MPs compared to sVEGF groups. Cell aggregates containing VEGF-MPs were also less susceptible to apoptosis. After 12 h (but not at 24 h) under hypoxia, caspase 9 activity was lower for the cell aggregates exposed to sVEGF or containing VEGF-MPs than aggregates containing blank MPs (Fig. 2d). Moreover, TEM results at 24 h

confirmed that OEPC aggregates containing VEGF-MPs showed less stress- and cell death-related lipid droplets[15] (Supplementary Fig. 4b–d) and lysosomes[16] (Supplementary Fig. 4e) than the ones containing blank MPs, sVEGF, or MPs (Supplementary Fig. 4b–e).

VEGF-MPs also enhanced the stabilization of vascular networks formed by OEPCs when cultured on top of Matrigel. To evaluate the angiogenic potential of OEPC aggregates with and without VEGF-MPs, cell aggregates were cultured on top of Matrigel under hypoxia (0.5% $O_2$). The Matrigel assay showed differences in the interaction of VEGF-MPs or blank MPs with OEPCs. Most of the VEGF-MPs were in close contact with OEPCs (Supplementary Fig. 5c, d), while a significant number of blank MPs were dispersed throughout well and not in the vicinity of sprouting OEPCs. The network length and branching points were then assessed. The highest network length and number of branch points were observed in cell aggregates exposed to sVEGF and VEGF-MPs (Fig. 2e, f). Importantly, the highest long-term stability of the networks was achieved when VEGF-MPs were used. After 60 h, the number of branch points decreased more than 50% for all the conditions, while the decrease was only 30% for cells activated by VEGF-MPs (Fig. 2f).

To demonstrate the in vivo survival effect of VEGF-MPs, cell aggregates containing $1 \times 10^6$ GFP-expressing OEPCs with uncoated MPs, VEGF-MPs, or sVEGF were mixed with a fibrin gel precursor solution and injected subcutaneously in mice. OEPC survival was monitored by an IVIS system (Fig. 3a). Animals treated with the OEPC aggregates containing blank MPs showed a rapid decrease in the fluorescence signal, which was lost at 3 day post-transplantation (Fig. 3a, b). Similarly, animals treated with OEPC aggregates containing sVEGF showed a rapid decrease in the fluorescence signal demonstrating a relatively poor cell survival. However, the animals treated with the OEPC aggregates containing VEGF-MPs showed a fluorescence signal until day 10 supporting the hypothesis that VEGF-MPs improve the survival of transplanted cells.

Overall, in vitro, immobilized VEGF enhanced OEPC survival in hypoxia conditions compared to sVEGF. This effect was characterized by a reduction in cell apoptosis as confirmed by a decrease in caspase 9 activity. Our results further indicate that immobilized VEGF has higher in vivo pro-survival effect than sVEGF in OEPC aggregates transplanted subcutaneously into nude mice.

**VEGF-MPs modulate *miR-17*, and *miR-217* expression.** The miRNA expression profile of OEPC aggregates was determined

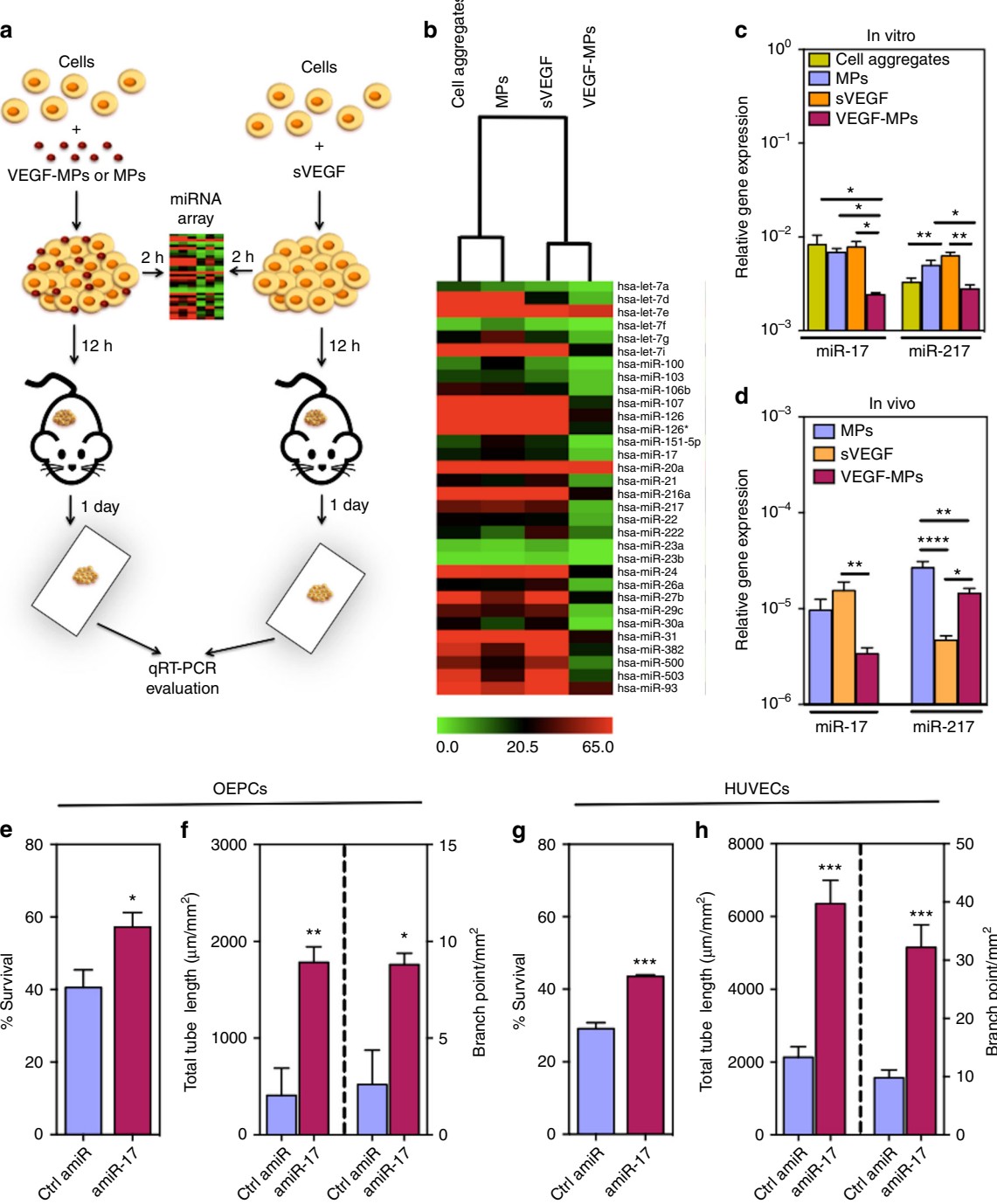

**Fig. 4** Identification of a miRNA associated with the function of OEPCs after contact with VEGF-MPs. **a** Schematic representation of the protocol used to identify miRNAs mediating the effect of VEGF-MPs. **b** Differentially regulated miRNAs ($P < 0.05$) in OEPC aggregates cultured in vitro for 2 h as evaluated by miRNA array. **c** Validation of some miRNAs by qRT-PCR. **d** miRNA expression as evaluated by qRT-PCR in OEPC aggregates implanted subcutaneously in mice for 1 day. U6 was used to normalize the data. In all graphs, values are given as average ± SEM ($n = 3$–4). Statistical analyses were performed using one-way ANOVA followed by a Bonferroni post test. *$P \leq 0.05$,**$P \leq 0.01$,***$P \leq 0.001$, and ****$P \leq 0.0001$. **e**, **g** Survival of OEPCs (**e**) or HUVECs (**g**) transfected with control antagomiR (Ctrl amiR) or *antagomiR-17* (*amiR-17*), in serum-deprived conditions for 48 h under hypoxia conditions (0.1% $O_2$), as assessed by Presto-Blue assay. **f**, **h** Transfected OEPCs or HUVECs with Ctrl amiR or *amiR-17* were cultured on Matrigel for 48 h under hypoxia after which the tube length and branching points were measured. In all graphs, values are given as average ± SEM ($n = 4$). An unpaired *t*-test was performed for statistical analysis between Ctrl amiR and *amir-17* groups. *$P \leq 0.05$, **$P \leq 0.01$, and ***$P \leq 0.001$

following 2 h activation with uncoated MPs, VEGF-MPs, or sVEGF by microarray (Fig. 4a). We further analyzed miRNAs that exhibited a significant expression difference ($P < 0.05$; *t*-test was performed using MeV software[28]) in OEPC aggregates containing VEGF-MPs relatively to the other groups (including

sVEGF) (Fig. 4b; Supplementary Fig. 6; Supplementary Data 1 and 2). Twenty miRNAs were downregulated in OEPC aggregates containing VEGF-MPs while one miRNA was upregulated (Supplementary Fig. 6). We initially focused our attention on *hsa-miR-217* and *hsa-miR-17*, because: (1) they were among the most

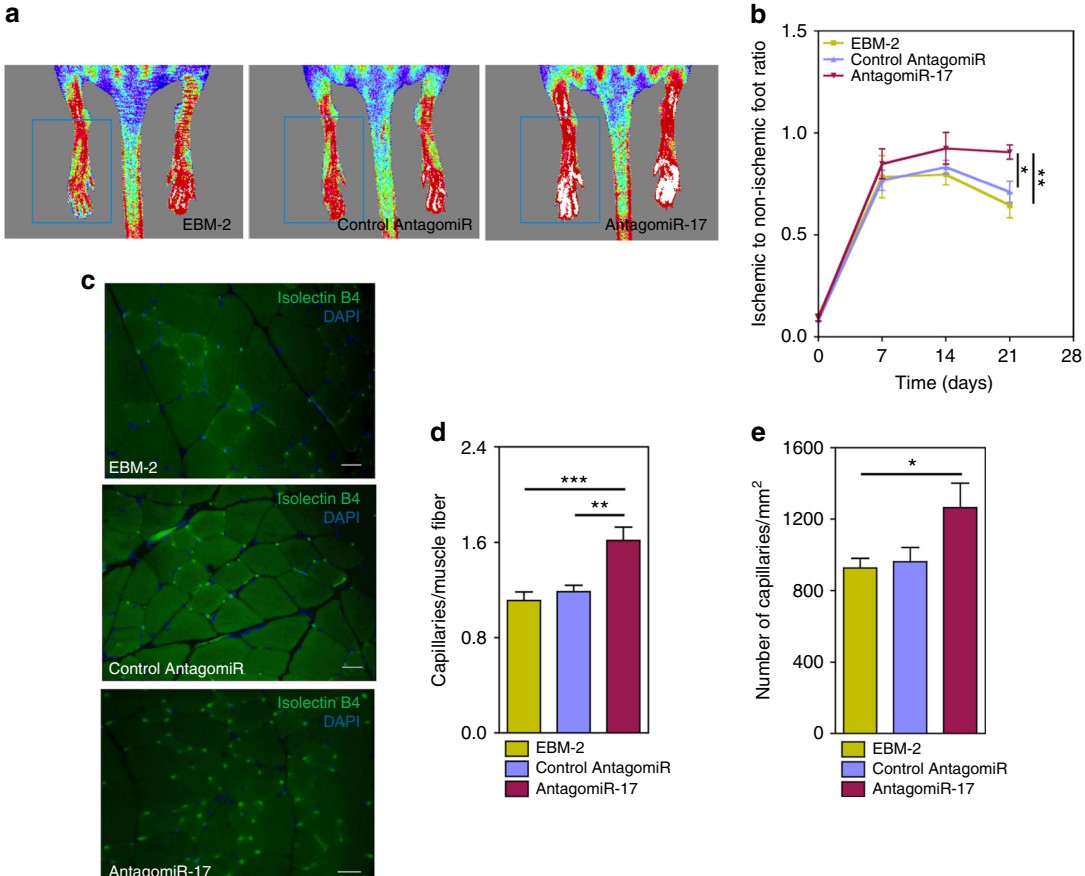

**Fig. 5** *AntagomiR-17*-treated OEPCs increase blood flow recovery and capillary density in a mouse model of hindlimb ischemia. **a**, **b** Unilateral limb ischemia was induced in female CD1 nude mice by occlusion of the left femoral artery. Immediately after occlusion, the ischemic muscles were injected with 3 million *antagomiR-17*-transfected OEPCs or 3 million control antagomiR-transfected OEPCs. Control group received only EBM-2. Blood flow recovery was measured using high-resolution laser color Doppler imaging and calculated (n = 12 mice/experimental group) as a ratio of ischemic over contralateral foot blood flow. **c–e** 21 days after surgery, limb muscles were harvested and prepared for immunohistochemical analyses. Capillary density in the adductor muscle was measured by staining with isolectin B4 (revealing endothelial cells) (**c**). The relative amount of positive cells was counted in eight randomly selected high-power fields (magnification ×20). *Scale bars*, 20 μm. (**d**, **e**). Data were shown as mean ± SEM. One-way ANOVA followed by a Bonferroni post test was used for statistical analysis. *$P \leq 0.05$,**$P \leq 0.01$, and ***$P \leq 0.001$

downregulated miRNAs in cell aggregates containing VEGF-MPs relatively to cell aggregates containing sVEGF (Supplementary Data 1), (2) downregulation of *hsa-miR-217* and *hsa-miR-17* has been associated with the prevention of vascular ageing[29] and enhancement of angiogenic activity[30], (3) qRT-PCR results confirmed that miR-17 and miR-217 were significantly downregulated when the VEGF-MPs were incubated with cells for a short time (2 h) (Fig. 4c).

Next, we investigated whether a similar trend in the expression of *hsa-miR-217* and *hsa-miR-17* could be found in vivo. Cell aggregates containing $1 \times 10^6$ OEPCs were subcutaneously injected in mice and retrieved 1 day after the surgeries (Fig. 4a). qRT-PCR results showed downregulation of *miR-217* expression in both sVEGF and VEGF-MP treated OEPC aggregates compared with OEPC aggregates treated with uncoated MPs. However, *miR-17* expression was uniquely downregulated only in the VEGF-MP treated group (Fig. 4d). Altogether, our results suggest that *miR-17* downregulation might be involved in the pro-survival effect of VEGF-MPs in OEPCs. For this reason, we focused our further studies on *miR-17*.

**AntagomiR-17 increases OEPC survival and angiogenesis.** In order to mimic the downregulation of *miR-17* by immobilized

VEGF, OEPCs were transfected with *antagomiR-17* (*amiR-17)* using Lipofectamine® RNAiMAX and OEPCs survival was evaluated after 48 h in hypoxic conditions (0.1% O₂). As controls, we used human umbilical vein ECs (HUVECs) to understand whether the effect of *amiR-17* was specific to OEPCs or it could be a broader pro-survival molecule for both progenitor and mature ECs, thus increasing the therapeutic potential of VEGF-MPs. *miR-17* downregulation after *amiR-17* transfection in OEPCs and HUVECs was confirmed by qRT-PCR (Supplementary Fig. 7a). Cell viability assay showed that *amiR-17* increased the survival of both cell types in hypoxia (Fig. 4e, g). Moreover, *amiR-17* increased the angiogenic responses in both OEPCs and HUVECs under hypoxic conditions compared to control amiR-treated groups (Fig. 4f, h). Next, we investigated whether the aforementioned positive effects of *miR-17* inhibition in OEPCs were relevant for the therapeutic performance of the cells after their transplantation in mouse ischemic limbs. Indeed, pre-treatment of OEPCs with *amiR-17* before transplantation accelerated the post-ischemic hemodynamic recovery (Fig. 5) and increased the capillary density of ischemic limb 21 days after the surgery (Fig. 5c–e).

The prevalent function of a miRNA is to inhibit the translation of a series of mRNA targets (usually called miRNA target

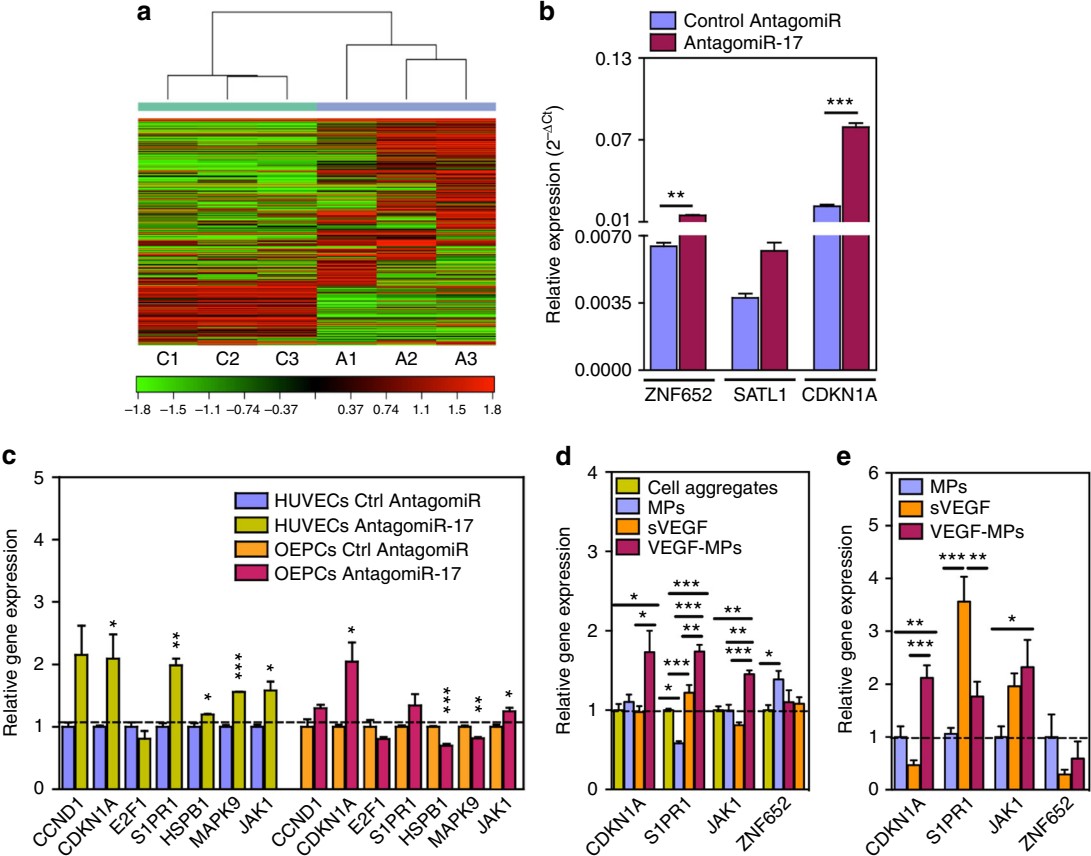

**Fig. 6** Gene targets of *miR-17* in OEPCs. **a**, **b** mRNA sequencing was performed for OEPCs transfected with control antagomiR or *antagomiR-17* for 48 h. **a** The heat map diagram showing the result of the two-way hierarchical clustering of RNA transcripts and samples, by including the top 500 transcripts (genes) that have the largest log2 fold difference based on FPKM counts. *Each row* represents one RNA transcript and *each column* represents one sample. The *color* of each point represents the relative expression level of a transcript across all samples. The *color scale* is shown at the *bottom*: *red* represents an expression level above the mean; *green* represents an expression level below the mean. On heat map, C is control amiR while A is *amiR-17*. **b** Validation of three gene targets by qRT-PCR. **c** The expression of previously reported *miR-17* gene targets in HUVECs and OEPCs transfected with *amiR-17*. The results were normalized to control amiR group for each gene. Among the genes tested, *CDKN1A* was significantly upregulated in both of the cell types transfected with *amiR-17*. The upregulation in the expression of *CDKN1A* was also confirmed in OEPC aggregates containing conjugated VEGF in both 24 h in vitro (**d**) and 24 h in vivo (**e**) samples. The gene expression results were normalized to cell control group (cell aggregates). Results are average ± SEM (*n* = 4–8). In **b** and **c**, unpaired *t*-test was performed between ctrl amiR and *amir-17* groups, while one-way ANOVA followed by a Bonferroni post test was used among groups in **d** and **e** for statistical analysis. *$P \leq 0.05$, **$P \leq 0.01$, and ***$P \leq 0.001$

"genes"). To define the target genes of *amiR-17* in OEPCs, we used next generation mRNA sequencing (Fig. 6a; Supplementary Fig. 7b, c; Supplementary Tables 3–6). mRNAs that were upregulated by *amiR-17* were chosen as direct target genes of the *miR-17*. These included: (1) *ZNF652* (zinc finger protein 652), which has a tumor suppressive function[31], (2) *SATL1* (spermidine/spermine N1-acetyl transferase-like 1 protein), which has a role in the ubiquitination and degradation of HIF-1a which in turn has a critical role in angiogenesis[32], and (3) *CDKN1A* (cyclin-dependent kinase inhibitor 1 A, also known as p21), which has a critical role in cell survival[33] (Fig. 6b; Supplementary Fig. 7b, c). PCR-based validation confirmed that both *ZNF652* and *CDKN1A* transcripts were up-regulated in OEPCs transfected with *amiR-17* vs cells transfected with control amiR. We additionally analyzed previously validated *miR-17* targets, although our sequencing data did not show *amiR-17* to alter their expression (Fig. 6c). In both OEPCs and HUVECs, downregulation of *miR-17* by its inhibitor (*amiR-17*) increased the expression of several genes, including endothelial differentiation gene *S1PR1* and *JAK1*, which is important in vascular homeostasis. Next, we investigated whether these changes in gene expression driven by *amiR-17* in OEPC could be replicated by

treatment with VEGF-MPs. Indeed, as compared with either sVEGF or uncoated MP treated groups, VEGF-MPs upregulated *CDKN1A* in OEPC aggregates both in vitro and in vivo (Fig. 6d, e). We additionally confirmed the upregulation of CDKN1A protein in OEPCs following *amiR-17* treatment (Supplementary Fig. 8). Overall, these results suggest that VEGF-MPs increase the survival of OEPCs by downregulating *miR-17* and subsequently upregulating its target genes, particularly *CDKN1A*.

**CDKN1A and ZNF652 regulate pro-survival effect of *amiR-17*.** To confirm the direct binding of *miR-17* to *CDKN1A* and *ZNF652*, we used a luciferase assay to co-transfect cells with individual 3′-untranslated region (UTR)-reporter constructs along with a *miR-17* mimic (to overexpress *miR-17*) (Fig. 7a). We further tested the functions of *CDKN1A* and *ZNF652* in OEPCs. OEPCs were double-transfected with either *CDKN1A* small interfering RNA (siRNA), *ZNF652* siRNA, or control siRNA in combination with either *amiR-17* or control amiR. The silencing for each target was confirmed at mRNA level (Supplementary Fig. 9a) and for CDKN1A in protein level (Supplementary Fig. 9b). The treatment of OEPCs with siRNAs against these gene targets prevented the prosurvival effect

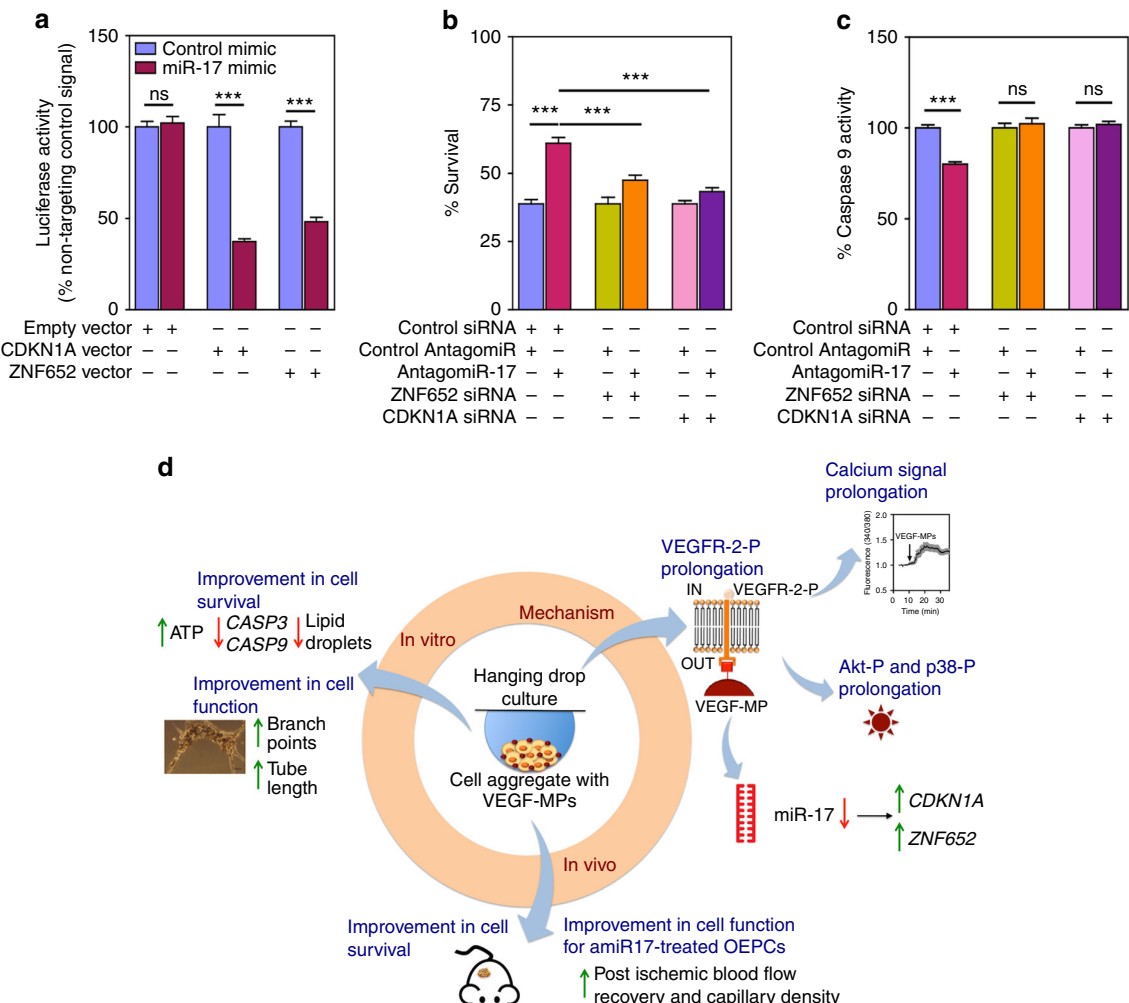

**Fig. 7** *AmiR-17* exerts its pro-survival effect by upregulating *CDKN1A*. **a** HEK293 cells were double transfected with 100 ng of each 3′UTR luciferase reporter vectors (empty vector, *CDKN1A* vector or *ZNF652* vector) and 50 nM miRNA mimics (control miRNA or *miR-17*). Signals from 3′UTR reporters for the *CDKN1A* and *ZNF652* were significantly knocked down when co-transfected with *miR-17*, but not with control miRNA. No difference between control miRNA and *miR-17* was observed when the empty vector was used. **b**, **c** OEPCs cultured in monoculture were silenced for *ZNF652* and *CDKN1A* by the use of siRNA. After 2 days of transfection, OEPCs were washed and the cell culture medium was replaced by EBM-2 and cells were incubated under hypoxia conditions (0.1% O$_2$), with 5% CO$_2$. After 48 h, both cell survival (by a Presto-Blue cell viability assay (**b**)) and cell apoptosis (**c**) were evaluated. In **b** and **c**, the values are given as average ± SEM ($n = 10$). Statistical analyses were performed using unpaired *t*-test. *$P \leq 0.05$, **$P \leq 0.01$, and ***$P \leq 0.001$. **d** Schematic representation of the action mechanism of the VEGF-conjugated microparticles used in this study. The interaction of conjugated VEGF with VEGFR-2 prolongs the phosphorylation of the receptor, calcium signaling and Akt phosphorylation compared with soluble VEGF group. Conjugated VEGF also downregulates *miR-17*, which leads to the upregulation of *CDKN1A* expression. Activation of Akt and downregulation of *miR-17* lead to an increase in cell survival by reducing apoptosis and favor sprout formation

that *amiR-17* had on OEPC exposed to hypoxia (Fig. 7b, c). In line with these functional data, the gene expression levels of the pro-apoptotic *CASP3* and *CASP9* were decreased by *amiR-17*, with this response being prevented by concomitant *CDKN1A* silencing (Supplementary Fig. 10a). Interestingly, OEPC aggregates containing VEGF-MPs also showed a decrease in these genes both in vitro and in vivo (Supplementary Fig. 10b, c). Moreover, *CDKN1A* siRNA, but not *ZNF652* siRNA, prevented the in vitro angiogenesis response to *amiR-17* (Supplementary Fig. 11).

## Discussion

In this study, we have investigated the pro-survival effect of immobilized VEGF in OEPCs. After demonstrating the bioactivity of immobilized VEGF by phosphorylation of

VEGFR-2 and induction of the intracellular accumulation of Ca$^{2+}$, we showed that immobilized VEGF enhanced the survival of OEPCs both in vitro and in vivo. We have further identified that these positive responses to immobilized VEGF in OEPCs are mediated by a decrease in *miR-17*, resulting in increased *CDKN1A* (p21) and *ZNF652* expressions (Fig. 7d).

We have used the VEGF-MPs to identify molecular targets mediating the prolonged pro-survival effect of immobilized VEGF in OEPCs. Our MP-based system has several advantages over planar surfaces with immobilized VEGF[13, 14, 34]. First, our MPs can be easily conjugated or manipulated by magnetic devices. Second, VEGF-MPs can be incorporated into cell aggregates for in vivo transplantation and the downstream targets of VEGF can be evaluated in an in vivo setting. Magnetic MPs were chosen in this study because they are easily controlled by a magnet facilitating the synthesis and purification of VEGF-MPs

and the characterization of cell aggregates containing VEGF-MPs (removal of the MPs from cell lysates in western blot and RNA isolation studies). To prevent potential toxic effects of these MPs due to iron release[22–24], we have used polystyrene-coated iron oxide MPs. Our results indicate no significant effect in OEPC viability after MP uptake. It was also reported that the injection of high doses of iron (3000 μmol Fe kg$^{-1}$; 168 mg Fe kg$^{-1}$) to rats and beagle dogs did not induce any acute or subacute toxicity[35]. In our study, less than 1 mg Fe kg$^{-1}$ mice was injected in animal studies.

Our results indicate that VEGF-MPs induce a prolonged phosphorylation of VEGFR-2 and maintain high intracellular levels of Ca$^{2+}$ as compared to sVEGF. It has been shown previously that the extended phosphorylation on planar surfaces with conjugated VEGF is mediated by the phosphorylation at the site Y1214 of the C-terminal tail of VEGFR-2[13]. Both matrix-bound and soluble VEGF activate several pathways at similar kinetics except for p38. Conjugated VEGF showed higher and prolonged activation kinetics for p38 compared with soluble VEGF[13]. These studies indicated that some pathways are preferentially selected according to the VEGF affinity to the matrix, VEGF presentation and the intracellular trafficking of VEGFR-2[13, 25]. However, due to the internalization of MPs by the OEPCs in our system, the prolonged pY1175 might activate survival pathways, such as Akt. As an expected result of prolonged activity, immobilized VEGF augmented the survival of OEPC aggregates in hypoxia.

Although some biological aspects of immobilized VEGF have been explored, no study targeted the use of immobilized VEGF to improve cell survival both in vitro and in vivo[13, 14]. Previous studies have shown that VEGF immobilized on different substrates including titanium, fibrin, and collagen is superior to sVEGF in promoting EC proliferation[36, 37] and EC branching[22]. In addition, a previous study has shown that VEGF physically immobilized to cell culture substrates could mediate EC survival after exposure to tumstatin, a proapoptotic agent[38]. However, no side-by-side comparison of the prosurvival effect of immobilized VEGF and sVEGF has been done and no in vivo prosurvival effect of immobilized VEGF has been reported. In this work, we have demonstrated the prosurvival effect of immobilized VEGF in both in vitro and in vivo. Subcutaneous injections of cell aggregates containing VEGF-MPs revealed that immobilized VEGF is functional in vivo and it enhances cell survival to a greater extent than sVEGF.

We have investigated miRNAs as the molecular targets of immobilized VEGF. Our in vitro and in vivo results show that the dowregulation of miR-17 is important for enhanced OEPC and EC survival. miR-17 was shown to control cellular proliferation and apoptosis by targeting the E2F family of transcription factors[39, 40]. Individual members of miR-17-92a cluster, e.g., miR-17, reduced EC sprouting whereas inhibitors of these miRNAs augmented angiogenesis in vitro and in vivo by targeting JAK1[30]. The JAK/STAT signaling pathway plays a critical role in the vascular homeostasis and disease. Interestingly, it was shown that inhibition of miR-17 did not affect tumor angiogenesis, indicating a context-dependent regulation of angiogenesis by miR-17 in vivo[30]. Our results indicated that another target of miR-17, CDKN1A, was upregulated both in vitro and in vivo when amiR-17 or conjugated VEGF was used. Although CDKN1A is well-known as a cell-cycle inhibitor, it has diverse biological activities such as EC survival and migration[33, 41]. In the literature, it was shown that CDKN1A gene transfer prevents apoptosis both in vitro and in vivo, following the interruption of blood flow[33]. In line, we have showed that the silencing of CDKN1A using siRNA reduces amiR-17-mediated OEPC survival and angiogenesis by upregulating apoptosis-

related genes such as CASP3 and CASP9 and increasing caspase 9 activity. In summary, the present work suggests that immobilized VEGF might increase OEPC survival by downregulating miR-17 and subsequently upregulating CDKN1A and ZNF652, which are direct targets of miR-17.

The transcriptional activation of the miR-17-92a cluster upon sVEGF treatment was reported in the literature and the activation of this cluster as a unit was reported to induce proliferation and angiogenic sprouting[20, 42]. However, in these studies, the authors did not investigate the function of individual members of this cluster. It is known that miR-17-92a cluster has both pro-angiogenic and anti-angiogenic miRNAs and its regulation of angiogenesis is context dependent[30, 43]. These previous works also showed that the miR-17-92a cluster activation by sVEGF is time and cell dependent[44]. The results show no statistically significant change in the miRNA levels of miR-17-5p after 6 h and 12 h[44]. This might be the reason why sVEGF does not induce miR-17 expression in our system in vitro. It was also shown that VEGF stimulates the expression of miR-17-92a cluster in human macrovascular venous ECs as well as in mouse microvascular lung ECs, but not in arterial ECs[44]. Overall, context-, cell-, and time-dependent regulation of miR-17-92a cluster might explain the differences between our results and these studies.

In conclusion, we show that VEGF-MPs improve the survival and angiogenesis of OEPCs both in vitro and in vivo. Immobilized VEGF prolonged the VEGFR-2 phosphorylation and Akt signaling up to 1 h, which were diminished in 10 min when sVEGF was used. VEGF-MPs promoted OEPC survival up to 10 days in subcutaneous injections. Our work also reveals that miR-17 is an important molecular target of VEGF-MPs in OEPCs. The downregulation of miR-17 both in vitro and in vivo is associated with an up-regulation of CDKN1A and ZNF652 transcripts. Our study provides insights about the molecular mechanism of immobilized VEGF in terms of OEPC angiogenesis and survival.

## Methods

**Preparation of VEGF-conjugated magnetic MPs**. Histidine-tagged VEGF (his-VEGF; ProSpec-Tany TechnoGene Ltd., Ness-Ziona, Israel) was immobilized onto polystyrene-coated, streptavidin-bound magnetic MPs (CELLection Biotin Binder Kit, Thermo Fisher Scientific, Waltham, MA, USA) conjugated with biotinylated anti-histidine antibody (R&D Systems, Minneapolis, MN, USA; Supplementary Table 1). A schematic representation of the conjugation method is given in Fig. 1a. Briefly, biotinylated anti-histidine antibody (1.5 μg per 10$^6$ particles) solution in 0.1% (w/v) bovine serum albumin (BSA; Sigma-Aldrich, St. Louis, MO, USA) was added to a suspension of streptavidin-coated MPs at room temperature. After 30 min of interaction on a rotator, the MPs were removed from suspension by a magnet and washed three times with phosphate-buffered saline (PBS) containing 0.1% BSA. Then, the MPs were resuspended in his-VEGF solution (500 μl; 4.5 μg in 0.1% BSA per 10$^6$ MPs) for 30 min at room temperature on a rotator. Subsequently, the MPs were removed from suspension by a magnet and washed with PBS containing 0.1% BSA until no measurable leaching of VEGF was observed. The conjugated MPs were stored in PBS at 4 °C until use.

**Determination of the immobilized VEGF amount onto MPs**. In order to calculate the amount of his-VEGF immobilized onto magnetic MPs, a VEGF ELISA kit (PeproTech EC Ltd., Rocky Hill, NJ, USA) was used according to manufacturer's instructions. The amount of his-VEGF conjugated to MPs was obtained by subtracting the measured amount of his-VEGF in the washing solutions from the original amount of his-VEGF added to the MPs.

**Flow cytometry analysis of magnetic MPs**. Conjugation of VEGF to magnetic MPs was confirmed by flow cytometry. After the conjugation step, MPs were blocked with BSA (2% (w/v); Sigma-Aldrich) for 30 min at 25 °C, washed, and then stained with phycoerythrin (PE)-conjugated anti-human VEGF antibody (R&D Systems) for 45 min at room temperature, in the dark (Supplementary Table 1). Uncoated and anti-his antibody-coated particles (without VEGF) were used as controls. FACScan (BD Biosciences, San Diego, CA, USA) and BD Cell Quest Software (BD Biosciences, Franklin Lakes, NJ, USA) were used for the acquisition and analysis of the data.

**Determination of iron release from magnetic MPs.** ICP-MS was performed to determine the amount of Fe release from magnetic MPs. Briefly, MPs ($10^6$ MPs) were added into 1 ml 0.1% (w/v) BSA (Sigma-Aldrich) in PBS. Subsequently, the MPs were removed from suspension by a magnet and washed one more time with PBS containing 0.1% BSA. Then, cell culture medium (500 μl; EGM-2; Lonza, Gaithersburg, MD, USA) was added onto MPs and they were incubated at 37 °C up to 7 days. The release medium was collected from $n = 3$ samples at different time points (1, 2, 3, 5, and 7 days). MPs and cell culture medium were used as controls.

**Differentiation of CD34+ cells into OEPCs.** Human UCB sample collection was approved by the ethical committees of Dr. Daniel de Matos Maternity Hospital in Coimbra and Hospital Infante D. Pedro in Aveiro. Parents signed an informed consent form indicating the use of collected UCB in research, in compliance with Portuguese legislation. CD34+ cells were isolated from human UCB and differentiated into OEPCs[4, 31]. Briefly, CD34+ cells were cultured in EGM-2 medium (Lonza) supplemented with 20% (v/v) fetal bovine serum (FBS; Invitrogen, Carlsbad, CA, USA) and 50 ng ml$^{-1}$ of VEGF$_{165}$ (PeproTech Inc., Rocky Hill, NJ, USA), on 1% gelatin-coated 24-well plates ($2 \times 10^5$ cells per well). After 2–3 weeks of differentiation, cells were characterized by immunofluorescence to confirm their OEPC phenotype. For each experiment, the cells were expanded in 1% (w/v) gelatin-coated T75 flasks (BD Biosciences, Franklin Lakes, NJ, USA) in EGM-2 (Lonza) medium.

**Preparation of OEPC aggregates.** Confluent OEPC cultures (until passage 7) were detached and suspended in serum reduced EGM-2 (1% FBS; v/v) medium without VEGF that contains 20% (w/v) methocel[27, 45]. The methocel solution was prepared by dissolving carboxymethylcellulose (6 g) in EBM-2 basal medium (500 ml). After centrifugation, the gel-like supernatant was used for experiments. Cells suspended in methocel solution (20,000 cells per 30 μl of methocel solution) were seeded into nonadhesive bacteriological dishes (Greiner, Frickenhausen, Germany; in drops of 30 μl) and cultured at 37 °C (5% CO$_2$, 95% humidity) for 12 h. Under these conditions, suspended OEPCs formed spontaneous aggregates. After 12 h, cell aggregates were collected and transferred into untreated 384-well plates (Nunc, Penfield, NY, USA) containing fibrinogen solution. Fibrin gels were formed by crosslinking of fibrinogen in the presence of thrombin (both from Sigma-Aldrich). The fibrinogen solution was prepared by dissolving human fibrinogen (20 mg ml$^{-1}$) in Tris-buffered saline (TBS) (Sigma-Aldrich) at pH 7.4, and then sterilized by filtering through a 0.22 μm syringe filter (Acrodisc, Pall, NY, USA). Fresh thrombin solutions were prepared by dissolving human thrombin in TBS at pH 7.4 at a concentration of 50 U ml$^{-1}$. Fibrin gels (30 μl) were prepared by mixing three different components: fibrinogen (10 mg ml$^{-1}$), CaCl$_2$ (Merck, Kenilworth, NJ, USA; 2.5 mM) and thrombin (2 U ml$^{-1}$). This solution was allowed to jellify at 37 °C and 95% relative humidity. After the solidification of the gel, serum-reduced (1% FBS; v/v) EGM-2 medium without VEGF was added into each well, on the top of the fibrin gels. The cells were then incubated at 37 °C and 5% CO$_2$ under hypoxia conditions (0.5% O$_2$) for 24 h.

**Toxicity studies for MPs.** To evaluate the effect of MPs on cell proliferation, OEPCs were seeded on 1% gelatin-coated 96-well plates and cultured with different ratios of MPs (1:10, 1:25, 1:50, and 1:100; cell:MP ratios) up to 5 days and the kinetics of proliferation was evaluated using cell proliferation reagent WST-1 (Roche Diagnostics, Mannheim, Germany) according to the manufacturer's instructions. This monolayer model was chosen due to the fact that cells in spheroids do not proliferate. For viability tests, cell spheroids with different cell numbers and cell:bead ratios were prepared as described above. The viability of the cells was determined up to 5 days using the CellTiter-Glo® luminescent cell viability assay (Promega, Madison, WI, USA) according to manufacturer's instructions (as described below).

**TUNEL stainings.** Apoptotic cell death in cell aggregates and cell aggregates treated with blank MPs was determined using In Situ Cell Detection Kit according to the manufacturer's instructions (Roche, Mannheim, Germany). This kit is based on labeling of DNA strand breaks by terminal deoxynucleotidyl transferase (TdT) and detection of incorporated fluorescein labels in nucleotide polymers by fluorescence microscopy. Briefly, cell aggregates were prepared as described in previous sections. After 24 h in hanging drops, cells were seeded into gelatin-coated 24-well μ-Plates from IBIDI (Munich, Germany) and incubated 24 h under hypoxia conditions (0.1% O$_2$). After incubation, the media were removed; cells were washed twice with PBS and fixated with 4% paraformaldehyde (EMS, Hatfield, PA, USA) for 15–20 min at room temperature. Cells were rinsed with PBS and incubated in permeabilization solution (0.1% Triton X-100 in 0.1% sodium citrate) for 2 min on ice. After washing with PBS, 100 μl TUNEL reaction mixture was added onto samples and the plate was incubated in a humidified atmosphere for 60 min at 37 °C in the dark. Then, cells were washed and incubated with mouse VECAD anti-human antibody (Santa Cruz, Dallas, TX, USA Biotechnology) for 1 h at room temperature. Anti-mouse Alexa Fluor 555 antibody (Molecular Probes, Eugene, OR, USA) was used as secondary antibody. The nuclei of the cells were stained with 4′,6-diamidino-2-phenylindole (DAPI; Sigma-Aldrich). After the labeling, the cells were examined using LSM 710 laser scanning confocal microscope (Carl Zeiss, Oberkochen, Germany). Information on antibodies is given in Supplementary Table 1.

**VEGFR-2 phosphorylation studies.** OEPCs cultured on gelatin-coated plates were starved in EBM-2 medium without supplements for 20 h, detached, and finally suspended in EBM-2 medium containing MPs or VEGF-MPs at a ratio of 1:2 (cell: particle). The cell suspension was then seeded as hanging drops (20,000 cells per 10 μl of methocel solution; this cell concentration was chosen to improve cell-MP interaction for short times) in nonadhesive bacteriological dishes (Greiner Bio One Ltd., Gloucestershire, UK) and incubated at 37 °C and 5% CO$_2$ for up to 1 h. After the incubation, the petri dishes were washed with cold EBM-2 medium and cells were collected into pre-chilled centrifuge tubes. Cell suspensions were centrifuged at 1000 rpm for 3 min at 4 °C. The medium was removed immediately, cells were washed with ice-cold PBS containing sodium vanadate (0.2 mM) and cell lysis buffer (1 ml; 1% NP-40 alternative, 20 mM Tris (pH 8.0), 137 mM NaCl, 10% glycerol, 2 mM EDTA, 1 mM activated sodium orthovanadate, 10 mg ml$^{-1}$ aprotinin, 10 mg ml$^{-1}$ leupeptin) was added onto the cells. Samples were allowed to sit on ice for 15 min. Before use, the lysates were centrifuged at 2000$g$ for 5 min and the supernatants were transferred into clean test tubes.

In order to determine the phosphorylation of VEGFR-2 by soluble VEGF, OEPCs cultured on gelatin-coated plates were starved in EBM-2 medium without supplements for 20 h. After that time, cells were activated with the same amount of soluble VEGF as used for immobilized VEGF for up to 30 min. The cell culture medium was removed, cells were washed with ice-cold PBS containing 0.2 mM sodium vanadate, lyzed and analyzed as described above.

Cell lysates were evaluated using phosho and total VEGFR-2 ELISA kits (R&D Systems) and the ratio of phospho/total VEGFR-2 was calculated according to manufacturer's instructions. The results were normalized by the cell culture control (for soluble VEGF samples) or uncoated particles (for conjugated-VEGF samples) for each experiment.

For western blots, total protein concentration from the lysates was determined by BCA assay, and samples were treated with SDS-PAGE buffer (6× concentrated: 350 mM Tris, 10% (w/v) SDS, 30% (v/v) glycerol, 0.6 M DTT, 0.06% (w/v) bromophenol blue), boiled for 5 min at 95 °C, and stored at −20 °C until used for western blotting. Then, proteins (40 μg of total protein) were resolved in 15% SDS-PAGE and then transferred to polyvinylidene difluoride (PVDF) membranes with 0.45 μm pore size in the following conditions: 300 mA, 90 min at 4 °C in a solution containing 10 mM 3-[Cyclohexylamino]-1-propanesulfonic acid (CAPS) and 20% methanol, pH 11. Membranes were blocked in Tris buffer saline containing 5% low-fat milk and 0.1% Tween 20 (Sigma-Aldrich) for 1 h at RT and then incubated overnight at 4 °C with the primary antibodies (rabbit anti-Tyr1175 (Santa Cruz, Dallas, TX, USA), rabbit anti-Tyr1214 (Santa Cruz) or mouse anti-VEGFR-2 (Abcam, Cambridge, UK)) diluted 100 times in 1% TBS-Tween and 0.5% low-fat milk. After rinsing three times with TBS-T 0.5% low-fat milk, membranes were incubated for 1 h at RT, with an alkaline phosphatase-linked secondary anti-rabbit or anti-mouse antibody 1:20,000 in 1% TBS-T and 0.5% low-fat milk (GE Healthcare, Buckinghamshire, UK). Protein immunoreactive bands were visualized in a Versa-Doc Imaging System (model 3000, BioRad Laboratories, CA), following incubation of the membrane with ECF reagent (GE Healthcare) for 5 min. Densitometric analyses were performed by using the ImageQuant software. Uncropped gel images are given in Supplementary Fig. 12.

**Akt and p38 phosphorylation studies.** The OEPC aggregates with soluble VEGF and VEGF-conjugated MPs were prepared as described on "VEGFR-2 phosphorylation studies" section with modifications. Briefly, OEPCs cultured on gelatin-coated plates were starved in EBM-2 medium without supplements for 24 h, detached by using TrypLE$^{TM}$ Express (Invitrogen), and suspended in EBM-2 medium containing soluble VEGF or VEGF-conjugated MPs at a ratio of 1:1 (cell:particle). The cell suspension was then seeded as hanging drops in non-adhesive bacteriological dishes (Greiner Bio One Ltd.) and incubated at 37 °C and 5% CO$_2$ for up to 1 h. After the incubation, the petri dishes were washed with cold EBM-2 medium and cells were collected into pre-chilled centrifuge tubes. Cell suspensions were centrifuged at 1000 rpm for 3 min at 4 °C. The medium was removed immediately, cells were washed with ice-cold PBS containing sodium vanadate (0.2 mM) and RIPA cell lysis buffer (Enzo, Farmingdale, NY, USA; 50 mM TRIS-hydrogen chloride, pH 7.4, containing 150 mM sodium chloride, 1 mM EDTA, 1 mM EGTA, 1% Triton X-100, 1% sodium deoxycholate and 0.1% SDS) containing protease inhibitor cocktail (Sigma-Aldrich) and 1 mM PMSF was added onto the cells. Samples were flash-frozen in liquid nitrogen. Before use, the lysates were centrifuged at 2000 $g$ for 5 min and the supernatants were transferred into clean test tubes. Three million cells were used per condition. Cell lysates were evaluated using phosho and total p38 and Akt ELISA kits (Invitrogen and Enzo) according to the manufacturer's instructions.

**Ultrastructural analysis of cell aggregates.** OEPC aggregates were fixed in 2.5% glutaraldehyde buffered with 0.05 M sodium cacodylate, postfixed in 1.0% osmium tetroxide and dehydrated in a graded series of ethanol[27, 46]. Micrographs were taken with a ZEISS EM-10 electron microscope at 80 kV.

**Single cell calcium measurements**. OEPCs growing in T75 flasks were detached and transferred onto 1% (w/v) gelatin-coated 10 mm coverslips. Cells were incubated at 37 °C and 5% $CO_2$ for 1 day, after which the cell culture medium was removed and cells were starved in M200 medium without supplements for 20 h. After removal of the cell culture medium, cells were loaded for 40 min at 37 °C with 5 μM Fura-2 AM (Invitrogen, Carlsbad, CA, USA), 0.1% (w/v) fatty acid-free BSA, and 0.02% (w/v) pluronic acid F-127 in Krebs buffer (132 mM NaCl, 1 mM KCl, 1 mM $MgCl_2$, 2.5 mM $CaCl_2$, 10 mM glucose, 10 mM HEPES, pH 7.4). In VEGFR-2 inhibition experiments, cells were co-incubated with Vatalanib (100 nM; SelleckChem, Kirby Drive, Houston, TX, USA). After a 30 min post-loading period at room temperature (RT), the coverslip was mounted on RC-25 chamber in a PH3 platform (Warner Instruments, Hamden, CT, USA) on the stage of an inverted Axiovert 200 fluorescence microscope (Carl Zeiss)[47]. Cells (~10 cells per field) were perfused with Krebs and stimulated by applying 100 μM histamine, 500 ng $ml^{-1}$ soluble VEGF (100 ng in 200 μl Krebs solution), $2 \times 10^5$ uncoated particles or $2 \times 10^5$ VEGF-coated MPs (in 200 μl Krebs solution) by a fast pressurized (95% air, 5% $CO_2$ atmosphere) system (AutoMate Scientific Inc., Berkeley, CA, USA).

$[Ca^{2+}]_i$ was evaluated by quantifying the ratio of the fluorescence emitted at 510 nm following alternate excitation (750 ms) at 340 and 380 nm, using a Lambda DG4 apparatus (Sutter Instrument, Novato, CA, USA) and a 510 nm long-pass filter (Carl Zeiss) before fluorescence acquisition with a 40× objective and a CoolSNAP digital camera (Roper Scientific, Tucson, AZ, USA). Acquired values were processed using the MetaFluor software (Universal Imaging, Downingtown, PA, USA).

**Determination of ATP production and caspase 9 activity**. Cellular viability was determined using the CellTiter-Glo® luminescent cell viability assay (Promega, Madison, WI, USA) according to manufacturer's instructions. In this assay, the luminescence signal increases with the number of viable cells in culture. The assay is based on the ATP present in the culture, which indicates the presence of metabolically active cells. The OEPC aggregates were prepared as described in previous sections. Equal volumes of CellTiter-Glo® reagent and cell culture medium were mixed in each well and incubated at room temperature for 2 min on an orbital shaker to induce cell lysis. Then, the cells were transferred into opaque-walled multiwell plates and luminescence measurements were done using a LUMIstar Luminometer (BMG LABTECH, Ortenberg, Germany). The medium without cells was used as a blank control.

The amount of apoptotic cells was determined using Caspase-Glo® 9 assay (Promega). The assay depends on the generation of a "glow-type" luminescent signal produced by the luciferase reaction between luminogenic caspase-9 substrate and caspase-9 enzyme present in the cells. The assay was performed as described previously, in "CellTiter-Glo® luminescent cell viability assay" part.

**Matrigel assay**. Angiogenesis μ-slides (IBIDI GmbH, Munich, Germany) were coated with Matrigel (10 μl per well; BD Biosciences, San Diego, CA, USA) and incubated for 30 min at 37 °C. The OEPCs growing on T75 flask in EGM-2 medium were detached, suspended in serum-reduced (1% FBS) EGM-2 medium without VEGF (50 μl; $1 \times 10^5$ cells $ml^{-1}$) and seeded on top of polymerized Matrigel. For the conditions with unconjugated and VEGF-conjugated MPs, the OEPCs were mixed with MPs at a ratio of 1:1 (cell:particle) and the cell-particle suspension was seeded on top of polymerized Matrigel after 5–10 min incubation to promote cell-particle interaction. For soluble VEGF condition, 50 ng $ml^{-1}$ VEGF was used. Cells were incubated under hypoxia conditions (0.5% $O_2$), with 5% $CO_2$. Sprout formation was evaluated by phase contrast microscopy (Zeiss Axiovert 40C, Carl Zeiss) 12 and 60 h after the cell seeding.

**VEGFR-2 stainings**. For VEGFR-2 stainings, 10,000 OEPCs per well were seeded on gelatin-coated 96-well μPlates from IBIDI (Munich, Germany). After overnight incubation at 37 °C in 5% $CO_2$, the cells were treated with VEGF-conjugated MPs (1:2, cell:bead ratio) or with same amount of soluble VEGF for different time points (i.e., 10, 30, 60, and 120 min). After incubations, the media were removed; cells were washed with PBS and fixated with 4% paraformaldehyde (EMS) for 15–20 min at room temperature. After permeabilizing the cells with 0.1% (v/v) Triton X-100 (Sigma-Aldrich) for 10 min, and blocking for 30 min with 1% (w/v) BSA solution (Sigma-Aldrich), the cells were stained for 1 h with the primary mouse VEGFR-2 (Abcam) and rabbit ZO-1 (Carlsbad, CA, USA) anti-human antibodies. Anti-rabbit Alexa Fluor 488 and anti-mouse Alexa Fluor 555 antibodies (Molecular Probes, Eugene, OR, USA) were used as secondary antibodies. The nuclei of the cells were stained with DAPI (Sigma-Aldrich). After the labeling, the cells were examined using LSM 710 laser scanning confocal microscope (Carl Zeiss). Information about antibodies is given on Supplementary Table 1.

**In vivo subcutaneous transplantation**. The Experimental Animal Committee of MIT approved all animal procedures. OEPC aggregates containing GFP-expressing OEPCs ($1 \times 10^6$; Passage 5; Angio-Proteomie, Boston, MA, USA) with or without $1 \times 10^6$ MPs were harvested, washed with EC basal medium, centrifuged and mixed in 200 μl fibrinogen (final concentration 10 mg $ml^{-1}$). Thrombin (final concentration 2 U $ml^{-1}$) was added to the mixture and the solution rapidly injected subcutaneously on lateral to the abdominal midline region of nude mice. Mice were

imaged under isoflurane anesthesia, by an IVIS® Spectrum in vivo imaging system (Xenogen Corporation, Alameda, CA, USA) up to 20 days. IVIS images were taken and analyzed using Caliper Living Imaging Software. The following parameters have been used: (i) a laser excitation at 490 nm and an emission filter at 510 nm, (ii) an exposure time of 0.5 s per image, and (iii) an image field of $12.5 \times 12.5$ cm. For miRNA isolations, animals were sacrificed 1 day after the surgery and the constructs were collected. The samples were washed with PBS, mounted in OCT compound (VWR, Radnor, PA, USA), and frozen in liquid nitrogen. Sections were cut at 20 μm on a Leica cryostat and immediately transferred to lysis buffer containing β-mercaptoethanol. The samples were disrupted with 5 mm stainless steel beads (Qiagen, Hilden, Germany) for 2 min at 25 Hz and for 2 min at 20 Hz using a Qiagen TissueLyser. miRNA was isolated from the samples as described in the following sections.

**miRNA microarray**. Total RNA was isolated using the absolutely RNA miRNA kit (400814, Agilent Technologies, Santa Clara, CA, USA) according to the manufacturer's instructions. RNA quality was assessed by an Agilent 2100 Bioanalyser (G2943CA), using an Agilent RNA 6000 Nano kit (5067–1511). Each RNA sample (100 ng) was hybridized with an Agilent human microRNA microarray (G4471A, Agilent Technologies). MicroRNA labeling, hybridization, and washing steps were carried out by following Agilent's instructions. Agilent microRNA assays integrate eight individual microarrays on a single glass slide. Each microarray includes ~15k features containing probes sourced from the miRBase public database. The probes are 60-mer oligonucleotides directly synthesized on the array. The human miRNA microarray version3 contains probes for 866 human and 89 human viral microRNAs from the Sanger miRBase v12.0[48].

The microarrays were scanned by an Agilent B Scanner (G2565BA). The raw data were analyzed using BRB-ArrayTools v3.4.0 developed by Dr. Richard Simon and BRB-ArrayTools development team[49]. This analysis generates a median normalized data set that is subjected to a statistical study and clustering using MeV software[28]. The differentially expressed miRNAs obtained from MeV were used to calculate the M-value and Fold-change variation. It was considered as differentially expressed miRNA a variation equal or higher than 2× between the different conditions. The data were uploaded on Gene Expression Omnibus (GEO Accession Number: GSE75899).

**mRNA sequencing**. Total RNA from OEPCs treated with *antagomiR-17* or control antagomiR for 48 h was isolated using mirVana™ miRNA isolation kit (Ambion, Carlsbad, CA, USA) according to the manufacturer's instructions. The RNA samples were shipped to Exiqon (Vedbaek, Denmark) in dry ice. RNA sample quality control, library preparation, library quality control and quantification and data analyses were performed by Exiqon (Vedbaek).

The library preparation was done using TruSeq® Stranded mRNA Sample preparation kit (Illumina Inc., San Diego, CA, USA). The starting material (500 ng) of total RNA was mRNA enriched using the oligodT bead system (manufacturer). The isolated mRNA was subsequently fragmented using enzymatic fragmentation. Then first strand synthesis and second strand synthesis were performed and the double stranded cDNA was purified (AMPure XP; Beckman Coulter, Brea, CA, USA). The cDNA was end repaired, 3′ adenylated and Illumina sequencing adaptors ligated onto the fragments ends, and the library was purified (AMPure XP). The mRNA stranded libraries were pre-amplified with PCR and purified (AMPure XP). The libraries' size distribution was validated and quality inspected on a Bioanalyzer high sensitivity DNA chip (Agilent Technologies). High quality libraries were quantified using qPCR, the concentration normalized and the samples pooled according to the project specification (number of reads). The library pool(s) were re-quantified with qPCR and optimal concentration of the library pool used to generate the clusters on the surface of a flowcell before sequencing on Nextseq500 instrument using High Output sequencing kit (50 cycles) according to the manufacturer instructions (Illumina Inc.). The data were uploaded on Gene Expression Omnibus (GEO Accession Number: GSE76663).

**Quantitative RT-PCR analyses of miRNA**. Total RNA was isolated using the absolutely RNA miRNA kit (400814; Agilent Technologies) according to the manufacturer's instructions. The cDNA was synthesized using the NCode™ miRNA first-strand cDNA synthesis kit (Invitrogen). The expression of selected miRNAs was assessed by quantitative RT-PCR (qRT-PCR) (7500 Fast Real-Time PCR System, Applied Biosystems, Carlsbad, CA, USA) using KAPA SYBR® FAST qPCR Master Mix (Kapa Biosystems, Wilmington, MA, USA). To normalize the expression levels of target miRNAs, the small nucleolar RNA C/D box 48 (SNORD48) and 5S ribosomal RNA were used as a control (housekeeping). Primer sequences are given in Supplementary Table 2.

**qRT-PCR analyses of total RNA in cell aggregates**. Total RNA was isolated from OEPCs and HUVECs under different conditions using mirVana™ miRNA isolation kit (Ambion) according to the manufacturer's instructions. In all cases, cDNA was prepared from 1 μg total RNA using Taqman Reverse transcription reagents (Applied Biosystems, Foster City, CA, USA). Quantitative real time PCR (qRT-PCR) was performed using KAPA SYBR FAST qPCR Master Mix (Kapa Biosystems) and the detection was carried out in a 7500 Fast Real-Time PCR

System (Applied Biosystems, Foster City, CA, USA). Quantification of target genes was performed relatively to the reference gene ($U6$): relative expression = $2^{[-(Ctsample-Cthousekeeping)]}$. Primer sequences are given as supporting information (Supplementary Table 2).

**Transfection of OEPCs.** OEPCs were transfected at 40–60% confluency using Lipofectamine RNAiMax (Invitrogen) according to the manufacturer's protocols. Inhibition of *miR-17* was achieved by transfection of cells with 50 nM miRIDIAN Hairpin Inhibitor (Dharmacon Inc., Lafayette, CO, USA). 50 nM anti-miR negative control from Ambion was used as control in all transfection experiments. siRNA-mediated silencing of *ZNF652* and *CDKN1A* was obtained using 40 nM ON-TARGETplus siRNAs (Dharmacon Inc.) against these targets. After 2 days of transfection, cells in 96-well plates were washed and the cell culture medium was replaced by EBM-2 and cells were incubated under normoxia and hypoxia conditions (0.1% $O_2$), with 5% $CO_2$. After 48 h, survival assay (Presto Blue Cell Viability Reagent; Thermo Fisher Scientific) was performed according to manufacturer's instructions. For Matrigel assay, the cells were detached after transfection, suspended in serum-reduced (1% FBS) EGM-2 medium without VEGF (50 μl; $2 \times 10^5$ cells ml$^{-1}$) and seeded on top of polymerized Matrigel. Cells were incubated under normoxia and hypoxia conditions (0.1% $O_2$), with 5% $CO_2$. Sprout formation was evaluated by phase contrast microscopy (Zeiss Axiovert 40C, Carl Zeiss) or INCELL Analyzer 2200 (GE Healthcare) 48 h after the cell seeding.

**Stainings for CDKN1A.** For stainings, the cells were detached after transfection, suspended in EGM-2 medium (50 μl; $5 \times 10^5$ cells ml$^{-1}$) and seeded onto gelatin-coated angiogenesis μ-slides (IBIDI GmbH, Munich, Germany). After 24 h in hypoxia, cells were fixated with 4% paraformaldehyde (EMS) for 15–20 min at room temperature. After permeabilizing the cells with 0.1% (v/v) Triton X-100 (Sigma-Aldrich) for 10 min, whenever required, and blocking for 30 min with 1% (w/v) BSA solution (Sigma-Aldrich), the cells were stained for 1 h with the primary rabbit CDKN1A (Santa Cruz, Dallas, TX, USA Biotechnology) anti-human antibody. The binding of primary antibody to specific cells was detected with anti-rabbit IgG Cy3 (Jackson Immunoresearch) conjugates. The nuclei of the cells were stained with DAPI (Sigma-Aldrich). After the labeling, the cells were examined with a fluorescence microscope (Carl Zeiss).

In order to determine the downregulation of CDKN1A in protein level after siRNA transfection, the cells were co-transfected with siRNAs and miRNAs in gelatin-coated 24-well μPlates from IBIDI (Munich, Germany) as described in the previous section. After 48 h of transfection, the medium was removed; cells were washed with PBS and fixated with 4% paraformaldehyde (EMS) for 15–20 min at room temperature. After permeabilizing the cells with 0.1% (v/v) Triton X-100 (Sigma-Aldrich) for 10 min, and blocking for 30 min with 1% (w/v) BSA solution (Sigma-Aldrich), the cells were stained for 1 h with the primary rabbit CDKN1A (Santa Cruz, Dallas, TX, USA Biotechnology) anti-human antibody. Anti-rabbit Alexa Fluor 488 antibody (Molecular Probes, Eugene, OR, USA) was used as secondary antibody. The nuclei of the cells were stained with DAPI (Sigma-Aldrich). After the labeling, the cells were examined using LSM 710 laser scanning confocal microscope (Carl Zeiss). Information about antibodies is given on Supplementary Table 1.

**Hindlimb ischemia experiments.** Experiments were performed in accordance with the Animal (Scientific Procedures) Act (UK) 1986 prepared by the Institute of Laboratory Animal Resources and under the auspices of UK Home Office Project and Personal License. Experiments were approved by the University of Bristol Ethical Review Committee. Before transplantation, the OEPCs were transfected with 50 nM control antagomiR (Ambion) or *antagomiR-17* (miRIDIAN Hairpin Inhibitor; Dharmacon Lafayette, CO, USA) for 48 h using Lipofectamine RNAiMax (Invitrogen) as described in the previous sections. Female CD1 nude mice (age 14 weeks) were injected with OEPCs ($3 \times 10^6$) transfected with *antagomiR-17* ($n = 12$), control antagomiR ($n = 12$), or EBM-2 ($n = 12$) after surgery to induce unilateral limb ischemia under general anesthesia. Limb ischemia was obtained by occlusion of the left femoral artery. Mice were sacrificed 3 weeks after surgery. The superficial blood flow to both feet was measured using high-resolution laser color Doppler imaging system at 30 min and days 2, 7, 14, and 21 after limb ischemia. Blood flow recovery was calculated as a ratio of ischemic over contralateral foot blood flow. After the last Doppler analysis (at day 21 after surgery), mice were perfusion-fixed under terminal anesthesia and limb muscles were harvested for immunohistochemical analyses.

**Histology.** The functional impact *antagomiR-17* treatment on CD1 nude mice was assessed by measuring capillary density in the adductor muscle. Eight micrometer thick muscle sections were stained using biotin-conjugated *Griffonia simplicifolia* isolectin B4 conjugate (I21414) and streptavidin-conjugated Alexa Fluor-488 (S11223; both from Molecular Probes, Eugene, OR, USA) to detect capillaries. Nuclei were stained with DAPI. The slides were mounted using Fluoromount-G™ (eBioscience, San Diego, CA, USA). The relative amount of positive cells was counted in 8 randomly selected high-power fields (magnification 20×) using a Zeiss inverted fluorescence microscope. Analyses were performed using muscles from eight mice per group. Capillary density was expressed in number per mm².

**3′UTR Luciferase assay.** 7000 HEK293 cells were seeded in each well of a 96-well black clear-bottom plate (Corning Inc., Corning, NY, USA) and incubated overnight at 37 °C (5% $CO_2$, 95% humidity). The cells were co-transfected with 50 nM miRNA mimics (control miRNA or miR-17; both from Ambion) and 100 ng of each 3′UTR luciferase reporter vectors (empty vector, *CDKN1A* vector or *ZNF652* vector; all from Active Motif, Carlsbad, CA, USA) using DharmaFECT Duo Transfection Reagent (Dharmacon Inc.)[50]. After 24 h of transfection, luciferase activity was measured using LightSwitch Luciferase Assay Reagents (SwitchGear Genomics, Menlo Park, CA, USA) according to the manufacturer's instructions. GloMax-96 Microplate Luminometer from Promega was used for the measurements.

**Statistical analysis.** An unpaired *t*-test or one-way analysis of variance (ANOVA) analysis of variance with Bonferroni post test were performed as statistical tests using GraphPad Prism software (San Diego, CA, USA, http://www.graphpad.com/). Results were considered significant when $P < 0.05$. Data are shown as mean ± SEM.

**Data availability.** The authors declare that all data supporting the findings of this study are available within the article and its supplementary information files or from the corresponding author upon reasonable request. All microarray and sequencing data have been deposited in the Gene Express Omnibus public database (National Center for Biotechnology Information) under accession codes: GSE75899 (miRNA microarray) and GSE76663 (mRNA sequencing).

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

## Acknowledgements

We acknowledge the assistance of Dr. Glenn Paradis (MIT, MA, USA) for flow cytometry analyses of the microparticles and Dr. Thomas Kraehenbuehl for his scientific advice. This work was funded by FEDER (Fundo Europeu de Desenvolvimento Regional) through the Program COMPETE and by Portuguese funds through FCT (Fundação para a Ciência e a Tecnologia) in the context of project PTDC/BIM-MED/1118/2012, and by the ERA Chair project ERA@UC (ref: 669088) through European Union's Horizon 2020 program. S.A. acknowledges doctoral and postdoctoral grants from FCT (SFRH/BD/42871/2008 and SFRH/BPD/105172/2014) and support from BHF (SS/CH/15/1/31199). C.E. is a BHF Professor in Cardiovascular Science. This study was supported by awards from Leducq Foundation Transatlantic Network on vascular microRNAs, MIRVAD (13 CVD 02) and BHF Regenerative Medicine Centers (RM/13/2/30158).

## Author contributions

S.A. designed and performed experiments, analyzed the data and wrote the manuscript. J.Z., M.B., L.C., J.S., R.F. and T.S. performed experiments and analyzed the data. L.B. read and corrected the manuscript. R.L. provided research funds, and corrected the manuscript. C.E. provided research funds, analyzed the data and rewrote parts of the manuscript. L.F. provided research funds, designed experiments, analyzed the data, co-wrote the manuscript. All authors approved the final manuscript.

## Additional information

**Competing interests:** The authors declare no competing financial interests.

