## [Peer Review File · Nature Communications]

Reviewer #1 (Remarks to the Author):

This study seeks to improve the longevity of transplanted endothelial progenitor cells. Authors elected to use an innovative approach, namely, presenting the cells with microparticle-carriers of VEGF-165, the strategy based on the known superiority of immobilized VEGF over the soluble one. They succeeded in generating such a platform technically and proceeded to confirm its effects in prolonged activation of VEGFR2 signaling. In vivo studies showed benefits of microparticle-VEGF-treated progenitor cells. In parallel, the investigators screened miRs affected by such a treatment and found that it induces miR-17, among others. By presenting endothelial progenitors with antagomiR-17 authors were able to mimic the effect of microparticle-VEGF therapy.

I find the evolution of this study pulsating and logical. Techniques involved appear to be adequate. The chances of future clinical applicability of described strategies are, in my opinion, high.

My only suggestion to authors is to consider the possibility that the effects of antagomiR could be accounted for by its ability to repress Wnt pathway, thus reducing endothelial-mesenchymal transition.

Michael S Goligorsky

Reviewer #2 (Remarks to the Author):

The approach of using microparticles for VEGF immobilization and the application in cell aggregates is interesting and the experiments were soundly carried out. Just one criticism is raised on the use of magnetic particles for cell aggregation. MPs are relatively toxic material, and not an optimal candidate for this cell aggregate mediator. The authors have to explain the direct effects of MPs (such as ionic release) on cells. Also detail how the cell/particle ratio is determined (Cell aggregate size, cell toxicity/viability, etc).

Reviewers' comments:

Reviewer #1 (Remarks to the Author):

This study seeks to improve the longevity of transplanted endothelial progenitor cells. Authors elected to use an innovative approach, namely, presenting the cells with microparticle-carriers of VEGF-165, the strategy based on the known superiority of immobilized VEGF over the soluble one. They succeeded in generating such a platform technically and proceeded to confirm its effects in prolonged activation of VEGFR2 signaling. In vivo studies showed benefits of microparticle-VEGF-treated progenitor cells. In parallel, the investigators screened miRs affected by such a treatment and found that it induces miR-17, among others. By presenting endothelial progenitors with antagomiR-17 authors were able to mimic the effect of microparticle-VEGF therapy. I find the evolution of this study pulsating and logical. Techniques involved appear to be adequate. The chances of future clinical applicability of described strategies are, in my opinion, high. My only suggestion to authors is to consider the possibility that the effects of antagomiR could be accounted for by its ability to repress Wnt pathway, thus reducing endothelial-mesenchymal transition.

The authors want to thank the reviewer for the positive comments and his valuable suggestion. As the reviewer pointed out, a relation between epithelial-mesenchymal transition (EMT) and miR-17-92 cluster has been previously described (Jiang *et al.*, Am J Pathol 2014, 184(5), 1335). It was reported that medium levels of miR-17-92 cluster induced PI3K/AKT by targeting *PTEN*, while high levels of miR-17-92 cluster inhibited the Wnt/ β -catenin pathway by targeting *CTNNB1* (Jiang *et al.*, Am J Pathol 2014, 184(5), 1355-1368). According to this paper, there was a significant decrease of multiple Wnt target genes downstream of β -catenin, including *AXIN2*, *CCND1*, *E2F1*, *KLF4*, *LEF1*, and *WNT3A* when miR-17-92 cluster levels were high. As a consequence of this, high levels of miR-17-92 cluster inhibited EMT by targeting Wnt/ β -catenin pathway. The authors of this work concluded that the inhibition of this pathway was especially related to the high levels of miR-18a. However, medium levels of miR-17-92 cluster activated PI3K/AKT by targeting *PTEN* and Wnt/ β -catenin pathway by targeting especially *CCND1* and *LEF1*, thus, increased EMT, which was correlated with high levels of EMT-related genes including *VIM*, *SNAIL1*, *SNAIL2*, *TWIST1*, *ZEB1* and *ZEB2*. *DABI* and *CELSR2* were also reported as miR-17 targets in Wnt pathway (Aydogdu *et al.*, Carcinogenesis, 2012, 33(8), 1502–1511).

Under the light of the reviewer comment, we focused on the Wnt and EMT-related genes in our sequencing data (GEO Accession Number: GSE76663). The results are given as tables below. Among >20 genes analyzed, just *CCND1* and *TCF4* were differentially expressed in Ctrl AmiR vs Amir-17 group (>2 fold changes). In addition, none of the EMT-related genes were differentially expressed (>2 fold changes) between these 2 groups. Therefore, according to our results, no apparent relation seems to exist between the inhibition of miR-17 and repression of Wnt pathway.

WNT-RELATED GENES

Gene	Ctrl Amir	AmiR-17	Log2 (FC)	p Value	q Value
DAB1,OMA1	11.7805	33.0795	1.48954	5.00E-05	0.000675141
C9,DAB2	86.579	119.021	0.459133	0.00725	0.0405763
WNT3A	0.0103057	0		1	1
CTNNB1	140.801	109.322	-0.365067	0.04895	0.165427
DVL1	12.0129	9.78656	-0.295711	0.1409	0.332688
DVL2	23.6415	13.7678	-0.78002	0.0018	0.0133201
LRP6	11.9355	8.34314	-0.516598	0.2147	0.410809
AXIN1	11.4031	10.4514	-0.125728	0.5863	0.761246
AXIN2,CEP112,CTD- 2535L24.2	22.0211	26.0651	0.243235	0.54515	0.731499
CCND1	174.086	32.0097	-2.44322	5.00E-05	0.000675141
E2F1	11.2698	10.65	-0.0816018	0.66965	0.816409
KLF4	0.939493	0.730975	-0.362059	0.3312	0.541577
LEF1	0	0.050387		1	1
CELSR2	0.464156	0.790325	0.767835	0.004	0.0253013
TCF4	133.07	29.1826	-2.189	5.00E-05	0.000675141
EGFR	0.964871	0.434585	-1.1507	0.0001	0.00125545
MYC	53.7506	59.9707	0.157977	0.35555	0.565109

EMT-RELATED GENES

Gene	Ctrl Amir	AmiR-17	Log2 (FC)	p Value	q Value
VIM	2538.28	4926.66	0.956755	5.00E-05	0.000675141
SNAI1	9.55636	11.4252	0.257684	0.19645	0.391951
SNAI2	0.57429	1.00658	0.809608	0.00505	0.0305166
TWIST1	0.0457495	0.0228323	-1.00268	1	1
ZEB1	35.2767	11.6101	-1.60334	5.00E-05	0.000675141
ZEB2	19.9293	7.93386	-1.3288	0.0002	0.00228787

Reviewer #2 (Remarks to the Author):

The approach of using microparticles for VEGF immobilization and the application in cell aggregates is interesting and the experiments were soundly carried out. Just one criticism is raised on the use of magnetic particles for cell aggregation. MPs are relatively toxic material, and not an optimal candidate for this cell aggregate mediator. The authors have to explain the direct effects of MPs (such as ionic release) on cells. Also detail how the cell/particle ratio is determined (Cell aggregate size, cell toxicity/viability, etc).

The authors have performed additional experiments to address the issue raised by the reviewer. The results are now presented in a new figure (Supplementary Fig. 2) and in the main manuscript on pages 9-10: “The toxicity of MPs against OEPCs was determined by cell proliferation, viability and apoptosis assays. In order to evaluate the effect of MPs on cell proliferation, monolayers of OEPCs were used (**Supplementary Fig. 2A**). The cells were treated with different ratios of MPs (1:10-1:100; cell:MP ratios) up to 5 days and cell proliferation quantified by a WTS-1 assay. No measurable effect of MPs on cell proliferation was observed (**Supplementary Fig. 2A**). Then, cell viability was monitored in cell aggregates having different cell numbers and cell:MP ratios (**Supplementary Figs. 2B and 2C**). Cell aggregates with small number of cells ($\leq 10,000$ cells) having a cell:MP ratio of 1:1 showed lower survival than cell aggregates without MPs (**Supplementary Fig. 2B**). Therefore, for subsequent experiments we have used cell aggregates with 20,000 cells having a diameter of approximately 150 μm (**Fig. 1A.3**). Then, we evaluated the cell:MP ratio. Cell aggregates with a cell:MP ratio of 1:1 seem to survive slightly better than the ones with a cell:MP ratio of 1:2 and thus were selected for subsequent experiments (**Supplementary Fig. 2C**). After selection of cell number and cell:MP ratio, we checked whether MPs increased apoptosis in cell aggregates. The MPs did not induce apoptosis in cell aggregates as determined by TUNEL assay (**Supplementary Fig. 2D**)”. In addition, the authors performed iron release studies (**Supplementary Fig. 1C**) to address the reviewer comment regarding the potential toxic effect of the iron release from MPs. Our release studies showed no significant Fe release from MPs (i.e. above the cell culture medium) up to 7 days at 37°C. This is likely due to the fact that the MPs, used in the entire work, were coated with polystyrene (this information was added in the revised version of the Supplementary Information, page 2). The authors have included information about how all these assays were performed in the “Supplementary Materials and Methods” section of the “Supplementary Information” file (pages 2-3; 5-6).

Authors have also discussed the potential toxic effects of MPs in the discussion section of the manuscript (page 18): “Magnetic MPs were chosen in this study because they are easily controlled by a magnet facilitating the synthesis and purification of VEGF-MPs and the characterization of cell aggregates containing VEGF-MPs (removal of the MPs from cell lysates in western blot and RNA isolation studies). To prevent potential toxic effects of these MPs due to iron release³²⁻³⁴, we have used polystyrene-coated iron oxide MPs. Our results indicate no significant effect in OEPC viability after MP uptake. It was also reported that the injection of high doses of iron (3000 $\mu\text{mol Fe/kg}$; 168 mg Fe/kg) to rats and beagle dogs did not induce any acute or subacute toxicity³⁵. In our study, less than 1 mg Fe/kg mice was injected in animal studies”.

Reviewer #1 (Remarks to the Author):

I'm satisfied with the authors' response to my suggestion

MSG

Reviewer #2 (Remarks to the Author):

The authors addressed the comments properly.